# Genome-wide association study of eosinophilic granulomatosis with polyangiitis reveals genomic loci stratified by ANCA status

Paul A Lyons et al.[#]

Eosinophilic granulomatosis with polyangiitis (EGPA) is a rare inflammatory disease of unknown cause. 30% of patients have anti-neutrophil cytoplasmic antibodies (ANCA) specific for myeloperoxidase (MPO). Here, we describe a genome-wide association study in 676 EGPA cases and 6809 controls, that identifies 4 EGPA-associated loci through conventional case-control analysis, and 4 additional associations through a conditional false discovery rate approach. Many variants are also associated with asthma and six are associated with eosinophil count in the general population. Through Mendelian randomisation, we show that a primary tendency to eosinophilia contributes to EGPA susceptibility. Stratification by ANCA reveals that EGPA comprises two genetically and clinically distinct syndromes. MPO+ ANCA EGPA is an eosinophilic autoimmune disease sharing certain clinical features and an *HLA-DQ* association with MPO+ ANCA-associated vasculitis, while ANCA-negative EGPA may instead have a mucosal/barrier dysfunction origin. Four candidate genes are targets of therapies in development, supporting their exploration in EGPA.

---

*email: kgcs2@cam.ac.uk  [#]A full list of authors and their affiliations appears at the end of the paper.

Eosinophilic granulomatosis with polyangiitis (EGPA), once named Churg-Strauss syndrome, has a unique combination of clinical features that have some overlap with the other anti-neutrophil cytoplasmic antibody (ANCA)-associated vasculitis (AAV) syndromes, granulomatosis with polyangiitis (GPA) and microscopic polyangiitis (MPA)[1,2]. The initial report by Churg and Strauss described necrotising vasculitis, eosinophilic tissue infiltration and extravascular granulomata at post-mortem[1]. After a prodromal period characterised by asthma and eosinophilia that may last for some years, patients develop the more distinctive clinical features of EGPA. These include various combinations of neuropathy, pulmonary infiltrates, myocarditis, and ear, nose and throat (ENT), skin, gastro-intestinal and renal involvement[2].

Despite being classified as a form of ANCA-associated vasculitis[3], vasculitis is not always evident (discussed in more detail in the Supplementary Note 1) and only 30–40% are ANCA-positive (almost all against myeloperoxidase (MPO) rather than proteinase-3 (PR3)). These observations, together with increasing evidence that clinical distinctions can be drawn between the ANCA-positive and ANCA-negative subsets of EGPA[4–6], suggest that clinically important subsets may exist within it[2]. The aetiology of EGPA is unknown; it is too uncommon for familial clustering to have been quantified or major genetic studies performed. Candidate gene studies in small cohorts have reported associations of EGPA with HLA-DRB4 and DRB1*07 and protection by DRB3 and DRB1*13[7,8], suggesting a genetic contribution of uncertain size.

Here we perform a genome-wide association study (GWAS) of EGPA. It demonstrates that EGPA is polygenic with genetic distinctions between MPO ANCA positive (MPO+) and ANCA-negative disease, correlating with different clinical features. The genetic associations themselves point to dysregulation of pathways controlling eosinophil biology, severe asthma and vasculitis, beginning to explain the development and clinical features of disease. These results suggest that EGPA might be comprised of two distinct diseases defined by ANCA status, and provide a scientific rationale for targeted therapy.

## Results

**The genetic contribution to EGPA.** We performed genome-wide association testing at 9.2 million genetic variants in 534 cases and 6688 controls. The EGPA patients' clinical features are summarised in Table 1. Further details of the cohorts are presented in Supplementary Tables 1–3. Consistent with previous reports[2],

176 (33%) were positive for ANCA, and of the 164 who were positive for specific ANCA by ELISA, 159 (97%) had MPO+ ANCA and five PR3 + ANCA. Genotyping was performed using the Affymetrix Axiom UK Biobank array and high-density genotype data was generated through imputation against the 1000 Genomes phase 3 reference panel (see the 'Methods' section). Despite attempting to control for population stratification by the inclusion of 20 genetic principal components (PCs) as covariates in the logistic regression model, the genomic inflation factor lambda remained elevated at 1.10, suggesting residual population stratification. We therefore used a linear mixed model (LMM; see the 'Methods' section), which more effectively controlled genomic inflation (lambda 1.047). Using this approach, we identified three genetic loci associated with EGPA at genome-wide significance ($P < 5 \times 10^{-8}$) (Fig. 1a, Supplementary Fig. 1, Table 2). The strongest association was with HLA-DQ, and the others were on chromosome 2 near BCL2L11 (encoding Bim) and on chromosome 5 near the TSLP gene (which encodes Thymic stromal lymphopoietin; TSLP). In addition, there was a suggestive association in an intergenic region at 10p14 ($P\ 8 \times 10^{-8}$).

To quantify the genetic influence on EGPA, the total narrow-sense heritability ($h^2$) was estimated: the genotyped variants additively explained ~22% of the total disease liability (EGPA $h^2 \geq 0.22$, standard error 0.082). Thus, while specific loci have substantial effect sizes in EGPA (Table 2), the contribution of genetics overall is similar to other immune-mediated diseases (e.g., $h^2$ estimates for Crohn's disease and ulcerative colitis are 26% and 19%, respectively[9]).

**Conditional false discovery rate analysis finds further loci.** One method to overcome limitations posed by GWAS sample size in rare diseases is to leverage results from GWAS of related phenotypes using the pleiotropy-informed Bayesian conditional false discovery rate (cFDR) method[10,11]. Asthma and eosinophil count were chosen as relevant traits, as they are both ubiquitous features of EGPA, and genetic variants showing association with them showed a trend for greater association with EGPA (Supplementary Fig. 2), with a consistent direction of effect (Supplementary Table 4), suggesting shared genetic architecture.

Use of the pleiotropy-informed Bayesian cFDR conditioning on asthma[12] revealed additional EGPA associations at 5q31.1 in C5orf56 (near IRF1 and IL5) and at 6q15 in BACH2. We then utilised data from a GWAS of eosinophil count in the general population[13], identifying further EGPA associations near LPP and in the CDK6 gene (Table 2).

---

**Table 1 Comparison of clinical features between MPO+ and ANCA−negative EGPA patients**

| | All patients n = 534 (%) | ANCA −ve n = 352 (%) | MPO+ ve n = 159 (%) | MPO+ ve vs. ANCA −ve P-value | Bonferroni-corrected P-value |
|---|---|---|---|---|---|
| Eosinophilia | 534 (100) | | | | |
| Asthma | 534 (100) | | | | |
| Neuropathy | 339 (63.5) | 201 (57.1) | 125 (78.6) | $4.5 \times 10^{-6}$ | **$3.6 \times 10^{-5}$** |
| Lung infiltrates | 301 (56.4) | 216 (61.4) | 72 (45.3) | 0.00098 | **0.0078** |
| ENT | 458 (85.8) | 309 (87.8) | 128 (80.5) | 0.042 | 0.34 |
| Cardiomyopathy | 135 (25.3) | 107 (30.4) | 23 (14.5) | 0.00020 | **0.0016** |
| Glomerulonephritis | 83 (15.5) | 33 (9.4) | 46 (28.9) | $3.2 \times 10^{-8}$ | **$2.6 \times 10^{-7}$** |
| Lung haemorrhage | 22 (4.1) | 14 (4.0) | 7 (4.4) | 1.0 | 1.0 |
| Purpura | 137 (25.7) | 91 (25.9) | 37 (23.3) | 0.60 | 1.0 |
| Positive biopsy* | 212 (41.3†) | 145 (42.9†) | 60 (38.5†) | 0.40 | 1.0 |

ENT ear, nose and throat. P-values were calculated from 1 degree of freedom chi-squared tests with Yates' continuity correction. Bonferroni correction was undertaken to account for the 8 clinical features tested; statistically significant P-values in bold
*Defined as a biopsy showing histopathological evidence of eosinophilic vasculitis, or perivascular eosinophilic infiltration, or eosinophil-rich granulomatous inflammation or extravascular eosinophils in a biopsy including an artery, arteriole, or venule
†Percentages are of those with available data. Biopsy data were unavailable for 21 (3.9%) of patients. Biopsy data was available for 156/159 (98.1%) MPO+ patients and 338/352 (96.0%) ANCA −negative patients

---

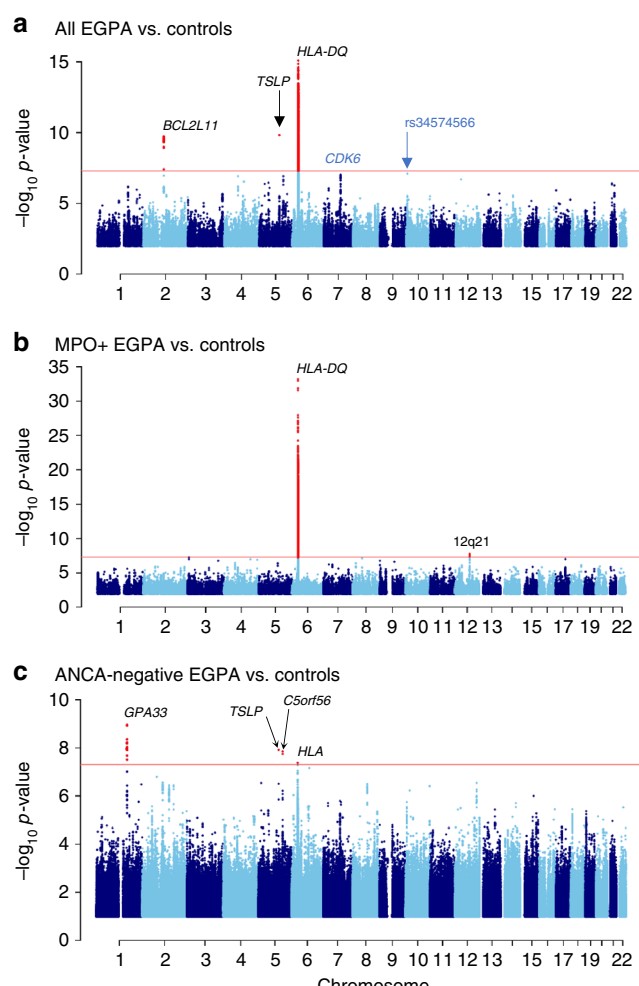

**Fig. 1** Manhattan plot of genetic associations with EGPA. Manhattan plots showing the association between genetic variants for (**a**) all EGPA cases ($n = 534$) vs. controls ($n = 6688$), (**b**) the subset of cases with MPO+ EGPA ($n = 159$) vs. controls, and (**c**) ANCA−negative EGPA cases ($n = 352$) vs. controls. Genetic variants at loci reaching genome-wide significance are highlighted in red. The red horizontal lines indicate the threshold for declaring genome-wide significance ($p = 5 \times 10^{-8}$). P-values for genetic association are from a linear mixed model (BOLT-LMM)

While replication is the gold standard for GWAS studies, the rarity of EGPA (annual incidence 1–2 cases per million) makes recruitment of an adequately powered replication cohort challenging. Nonetheless, a second cohort of 142 EGPA patients from two European centres was identified; 43 (30%) had MPO+ ANCA (Supplementary Table 5). There was a strong correlation between the estimated effect sizes in the primary and replication cohorts (Pearson $r = 0.96$, $p = 0.0002$), providing additional support for the reported associations (Supplementary Fig. 3 and Supplementary Table 6). Following meta-analysis of the primary and secondary cohorts, the signal at 10p14 became genome-wide significant ($P$ $2.9 \times 10^{-8}$). Thus in total we identified 8 associations with EGPA, consistent with it being a polygenic disease.

**Clinically and genetically distinct EGPA subsets.** EGPA has been classified alongside MPA and GPA as an AAV[3], despite ANCA being only found in the minority of cases. While asthma and eosinophilia are common to all patients with EGPA, the frequency of other clinical features of the disease vary between

**Table 2 Genetic associations with EGPA**

| Chr | Variant rsid | Gene/Region | Total EGPA N = 534 | | | | | MPO+ EGPA N = 159 | | ANCA −ve EGPA N = 352 | | cFDR asthma analysis N = 10,365 | | cFDR EC analysis N = 175,000 | |
|---|---|---|---|---|---|---|---|---|---|---|---|---|---|---|---|
| | | | Cont maf | Case maf | OR | Meta OR | Meta P | LMM P | MPO OR | MPO P | OR | P | P asthma | cFDR^ (EGPA\|asthma) | P EC | cFDR‡ (EGPA\|EC) |
| 2 | rs72946301 | BCL2L11 | 0.1 | 0.17 | 1.66 | 1.81 | **9.0 × 10⁻¹¹** | **1.9 × 10⁻¹⁰** | 1.89 | 7.7 × 10⁻⁵ | 1.76 | 3.6 × 10⁻⁷ | | | | |
| 5 | rs1837253 | TSLP | 0.26 | 0.17 | 1.42 | 1.52 | **5.2 × 10⁻¹¹** | **1.5 × 10⁻¹⁰** | 1.46 | 0.0008 | 1.53 | **1.2 × 10⁻⁸** | | | | |
| 6 | rs9274704 | HLA-DQ | 0.17 | 0.27 | 1.98 | 2.01 | **1.2 × 10⁻²⁰** | **8.2 × 10⁻¹⁶** | 5.68 | **1.1 × 10⁻²⁸** | 1.32 | 0.004 | | | | |
| 10 | rs34574566 | 10p14 | 0.31 | 0.24 | 0.73 | 0.7 | **2.9 × 10⁻⁸** | 8.0 × 10⁻⁸ | 0.66 | 0.0004 | 0.7 | 9.9 × 10⁻⁶ | | | 4.5 × 10⁻⁸ | **0.0003** |
| 7 | rs42041 | CDK6 | 0.24 | 0.31 | 1.32 | | | 1.9 × 10⁻⁶ | 1.34 | 0.014 | 1.36 | 9.7 × 10⁻⁵ | | | 1.6 × 10⁻²⁹ | **3.6 × 10⁻⁵** |
| 5 | rs11745587† | IRF1/IL5 | 0.35 | 0.4 | 1.31 | | | 2.1 × 10⁻⁷ | 1.16 | 0.17 | 1.47 | **1.8 × 10⁻⁸** | 0.002 | **1.9 × 10⁻⁵** | 1.0 × 10⁻¹⁸ | **0.0002** |
| 6 | rs6454802 | BACH2 | 0.4 | 0.31 | 0.8 | | | 2.2 × 10⁻⁶ | 0.81 | 0.024 | 0.74 | 3.8 × 10⁻⁶ | 0.002 | **0.0002** | 9.0 × 10⁻¹⁴ | **0.0002** |
| 3 | rs9290877 | LPP | 0.3 | 0.38 | 1.27 | | | 4.7 × 10⁻⁶ | 1.48 | 0.0007 | 1.24 | 0.0006 | | | | |
| 1 | rs72689399 | GPA33 | 0.01 | 0.03 | 2.7 | | | 6.7 × 10⁻⁷ | 0.89 | 0.96 | 5.34 | **1.1 × 10⁻⁹** | | | | |
| 6 | rs6931740 | HLA | 0.39 | 0.25 | 0.62 | | | **1.7 × 10⁻¹⁰** | 0.55 | 1.6 × 10⁻⁵ | 0.61 | **4.2 × 10⁻⁸** | | | | |
| 12 | rs78478398 | 12q21 | 0.03 | 0.05 | 0.59 | | | 0.0017 | 0.17 | **1.7 × 10⁻⁸** | 0.81 | 0.37 | | | | |

EC eosinophil count. Genome wide significant associations highlighted in bold. P asthma, p value from the GWAS by Moffatt et al.[12] P EC, p value from the GWAS by Astle et al.[13]
^Reached genome wide significance in EGPA by cFDR. Significance threshold cFDR < 3.9 × 10⁻⁴, FDR (excl MHC) < 4.3 × 10⁻³ (equivalent to P 5 × 10⁻⁸)
‡Reached genome wide significance in EGPA by cFDR. Significance threshold cFDR < 3.1 × 10⁻⁴, FDR (excl MHC) < 3.5 × 10⁻³ (equivalent to P 5 × 10⁻⁸)
§SNP most associated with EGPA that was directly genotyped in asthma. Stronger EGPA associations exist at this locus

ANCA-positive and ANCA-negative subgroups[2,4–6,14]. We confirmed these phenotypic differences, with glomerulonephritis and neuropathy (clinical features consistent with vasculitis) more prevalent in the MPO+ subgroup, whereas lung infiltrates and cardiac involvement were common in the ANCA-negative subgroup (Table 1). These differential clinical associations remained statistically significant after adjustment for country of origin, suggesting a true biological difference between the subgroups (see the 'Methods', section Supplementary Tables 7–8). These data, together with the analogous situation where patients with MPO+ ANCA or PR3 + ANCA GPA/MPA have distinct genetic associations[15,16], suggested that there may be genetic differences between MPO+ and ANCA-negative EGPA.

We therefore compared these subsets to healthy controls, excluding the PR3+ ANCA patients as there were too few to form a useful subset (Fig. 1b, c, Table 2). The analysis of MPO+ EGPA versus controls revealed an association at HLA-DQ (sentinel variant rs17212014; Supplementary Table 9), which was much stronger than in the comparison of all EGPA against controls, despite the reduced sample size in the former analysis (Table 2). A weaker association in the HLA region was also detected in the analysis of the ANCA-negative subset, but further examination of this signal revealed it was distinct from HLA-DQ (Supplementary Fig. 4). In addition, the MPO+ subgroup analysis revealed an association at rs78478398 on chromosome 12. Variants near GPA33 and at IRF1/IL5 reached genome-wide significance in the ANCA-negative subset, but were not associated with the MPO+ group (P 0.96 and 0.17, respectively, LMM). The associations at BCL2L11, TSLP, CDK6, BACH2, Chromosome 10, and LPP appeared to be independent of ANCA status, allowing for the reduced power in the smaller MPO+ subset (Table 2).

A signal that appears specific to one subgroup might reflect lack of power to detect it in the other rather than necessarily indicating true genetic heterogeneity between the two subgroups. To formally address this issue, we compared genotypes of MPO+ and ANCA-negative samples directly with one another (i.e., a within-cases analysis, independent of controls), and computed p-values for the 11 variants found to be significant in the previous analyses. A significant difference in allelic frequency between the two subtypes at a given genetic variant indicates a subtype dependent effect on disease susceptibility. This analysis of MPO+ vs. ANCA negative cases revealed a genome-wide significant association at rs17212014 in the HLA-DQ region (P $5.5 \times 10^{-13}$, logistic regression). No other signals were significant after Bonferroni adjustment for multiple testing (P value threshold 0.0045). 3 variants had unadjusted P-values < 0.05: rs11745587 in the C5orf56-IRF1-IL5 region (P 0.018), rs78478398 on chromosome 12 (P 0.02), and rs72689399 in GPA (P 0.0499; all p-values from logistic regression). Of note, the HLA association detected in the ANCA-negative vs. controls analysis was not evident in the within-cases analysis (P 0.19, logistic regression). In summary, this analysis provides robust evidence of a differential genetic basis of MPO+ and ANCA-negative EGPA at the HLA-DQ region.

In the small replication cohort, despite only 43 patients with MPO+ EGPA, the association at the HLA-DQ locus was replicated (Supplementary Table 9). No evidence of association was observed between variants at GPA33 and the ANCA-negative subgroup although, due to the low minor allele frequency and patient number, power was limited.

In light of the clinical and genetic differences identified between MPO+ and ANCA-negative EGPA, we tested whether one subset was more genetically similar to asthma than was the other (see the 'Methods' section, Supplementary Note 2). With the HLA region removed, ANCA-negative EGPA was more genetically similar to asthma than was MPO+ EGPA (Supplementary Note 2, Supplementary Figs. 5, 6). This provides evidence for genetic differences

between the two subtypes outside the HLA region, and suggests that the aetiology of ANCA-negative EGPA may more closely resemble that of asthma than does MPO+ EGPA.

Thus EGPA has a complex inheritance. Some loci are associated with both MPO+ ANCA and ANCA-negative EGPA, consistent with the phenotypic overlap between the subsets and with their shared prodrome. Others are associated with only one subgroup. That these genetically distinct subsets of EGPA align with clinical disease phenotype (Table 1) suggests that differences in pathogenesis exist and that the current clinical classification system should be re-visited (Fig. 3).

**Candidate genes and EGPA pathogenesis.** All genetic loci implicated in EGPA are detailed in Supplementary Figs. 7, 8, and four exemplars are shown in Fig. 2. We cross-referenced the lead EGPA-associated variant at each locus with disease-associated variants in linkage disequilibrium from the NHGRI GWAS Catalogue (Supplementary Data 1). It is interesting that 7 of the 8 alleles associated with increased EGPA risk are also associated with increased physiological eosinophil count at genome-wide significance (Fig. 4a, Supplementary Table 4). Given that some associations with EGPA were detected by cFDR analysis using eosinophil count GWAS data, we considered the possibility that this result was a consequence of our analytical approach. We therefore performed an unbiased assessment of all eosinophil-associated variants for their effect on EGPA risk in our primary GWAS analysis (i.e., agnostic of cFDR), and observed that the relationship between genetic effect on eosinophil count and EGPA risk held true (Supplementary Fig. 9). In addition, 5 non-MHC EGPA risk alleles also confer increased risk of asthma (Supplementary Table 4), and 2 confer increased risk of nasal polyps (Supplementary Table 10, Supplementary Data 1), consistent with the association between EGPA and these traits.

To identify potential candidate genes, we sought long-range interactions between the EGPA-associated variants and gene promoters and regulatory regions in promoter capture Hi-C datasets using the CHiCP browser (see the 'Methods' section). We emphasise that whilst Hi-C data can suggest a link between a disease-associated variant and a candidate gene, it does not provide conclusive evidence of causality. In addition, we identified genes for which the sentinel EGPA-associated variants (or their proxies in linkage disequilibrium (LD)) are expression quantitative loci (eQTLs) (Supplementary Data 2).

We sought external evidence from genomic databases (Supplementary Data 1–3) and the experimental literature (Supplementary Table 10) to provide corroboration for the candidate genes that we identified. For the majority of loci, we identified strong experimental evidence to implicate candidate genes in the pathogenesis of EGPA. Individual loci are discussed below.

**BCL2L11 and MORRBID.** The sentinel EGPA-associated variant lies in an intron in ACOXL, near BCL2L11, that encodes BIM, a Bcl2 family member essential for controlling apoptosis, immune homeostasis and autoimmune disease[17–19], and mast cell survival[20]. The EGPA-associated variant also lies within MIR4435-2HG (MORRBID), that encodes a long non-coding RNA that regulates Bim transcription, controls eosinophil apoptosis and may be dysregulated in hypereosinophilic syndrome[21]. In addition, Hi-C data showed interaction of the EGPA-associated variant with the promoter of MORRBID. These relevant functional associations suggest BCL2L11 and MORRBID are more likely than ACOXL to be the causal gene at this locus. The EGPA risk allele is also associated with higher eosinophil count, and with increased risk of asthma and primary sclerosing cholangitis (PSC)

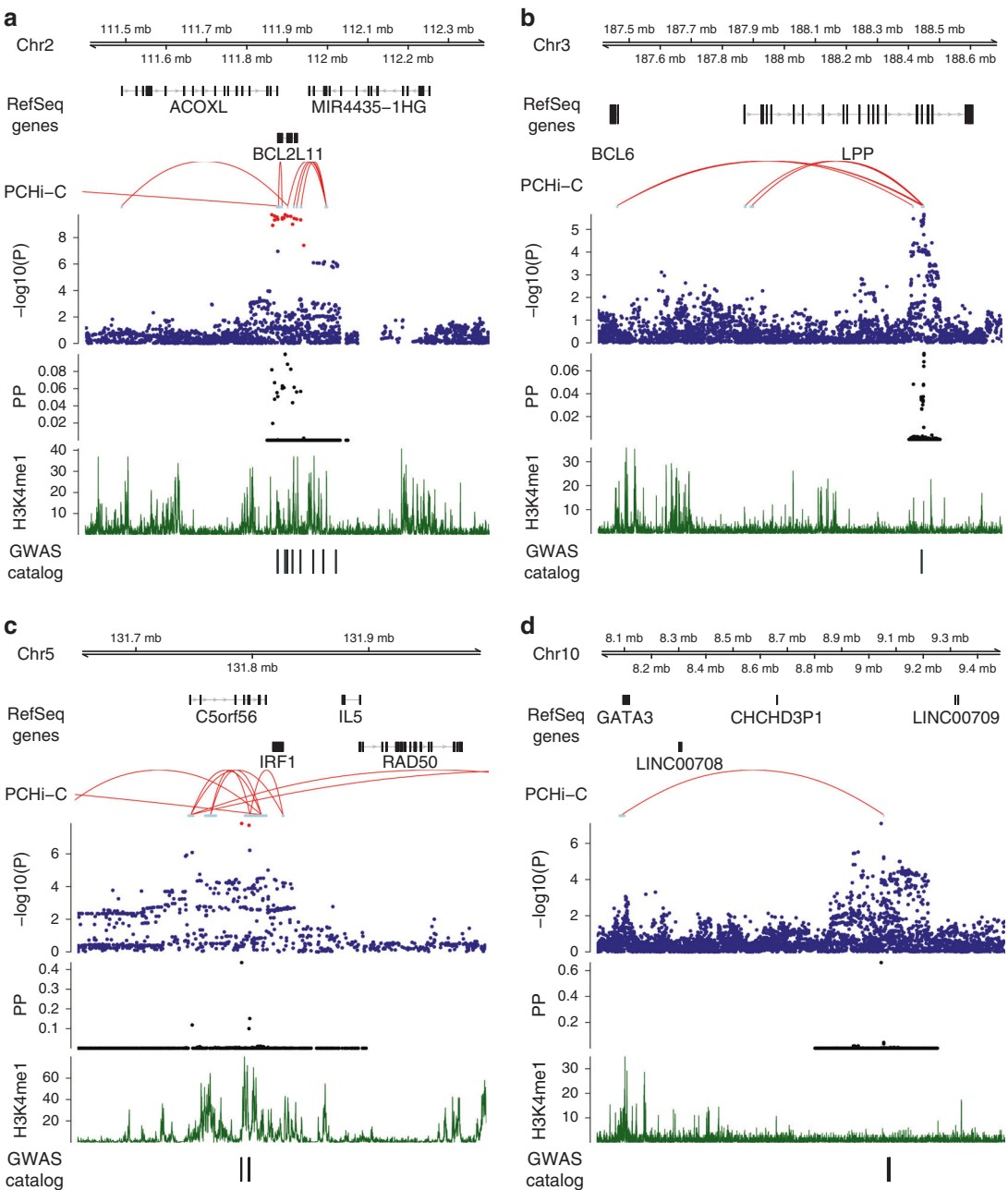

**Fig. 2** Genomic features at four EGPA-associated loci. Genomic positions from the hg19 genome build, representative RefSeq genes, long-range DNA interactions, genetic variant associations with EGPA, causal variant mapping (expressed as posterior probabilities, PP) and H3K4 mono-methylation data are shown for (**a**) the *BCL2L11* region, (**b**) the *LPP* region, (**c**) the *C5orf56-IRF1-IL5* region and (**d**) the 10p14 intergenic region. Arrows indicate direction of transcription. *P*-values for genetic association are from a linear mixed model (BOLT-LMM). For further details regarding the promoter enhancer interaction mapping, including cell types analysed at each locus, see the 'Methods' section and Supplementary Fig. 7

(Supplementary Table 10, Supplementary Data 1), diseases in which eosinophils have been implicated.

**TSLP**. The EGPA susceptibility variant rs1837253 lies immediately upstream of *TSLP*. TSLP is released by stromal and epithelial cells in response to inflammatory stimuli, and drives eosinophilia and enhanced TH2 responses through effects on mast cells, group 2 innate lymphoid cells (ILC2), and dendritic cells. The risk allele, rs1837253:C, is associated with higher TSLP protein secretion in stimulated nasal epithelial cultures[22]. No known genetic variants are in high LD with rs1837253, with no variants with $r^2 > 0.3$ in

European-ancestry populations in the 1000 Genomes phase 3 data, suggesting that it is either the causal variant or, alternatively, that rs1837253 is tagging a rare variant that was not present in the individuals sequenced in the 1000 Genomes Project. rs1837253:C is also associated with higher risk of asthma, nasal polyps, and allergic rhinitis and with higher eosinophil counts (Supplementary Table 10, Supplementary Data 1), increasing the risk of EGPA more strongly than it does asthma (OR 1.51 vs. 1.12–1.27: Supplementary Fig. 10). Other variants in the *TSLP* region, independent of rs1837253, are associated with asthma, eosinophil count, eosinophilic oesophagitis and allergic traits (Supplementary Data 3).

**GPA33**. *GPA33* encodes a cell surface glycoprotein that maintains barrier function in the intestinal epithelium[23]. Intriguingly, the EGPA-associated variant correlates with GPA33 expression in bronchial tissue[24], suggesting GPA33 control of respiratory or intestinal barrier function might play a role in EGPA pathogenesis. In keeping with this hypothesis, we find shared genetic architecture between ANCA-negative EGPA and inflammatory bowel disease (IBD) (Supplementary Fig. 11), a disease commonly attributed to mucosal barrier dysfunction. This association was not seen with MPO+ EGPA, which is not associated with the *GPA33* variant.

**LPP**. EGPA-associated rs9290877 is within *LPP*, encoding a LIM domain protein[25], and is associated with asthma, allergy and plasma IgE (Supplementary Table 10, Supplementary Data 1). CHiCP analysis links this variant to *BCL6* (Fig. 2b), encoding a transcriptional repressor central to immune, and in particular TH2, regulation; BCL6-deficient mice die of overwhelming eosinophilic inflammation characterised by myocarditis and pulmonary vasculitis[26].

**C5orf56-IRF1-IL5**. rs11745587:A is associated with increased susceptibility to EGPA, higher eosinophil count[13], asthma (including the severe asthma subtype) and allergic rhinitis, as well as IBD and juvenile idiopathic arthritis (Supplementary Table 10, Supplementary Data 1). CHiCP analysis shows interactions with the putative regulatory regions of *IL4, IL5,* and *IRF1*, all excellent candidates (Fig. 2c). IL-4 and IL5 are archetypal Th2 cytokines, and IL-5 in particular drives eosinophilic inflammation[27].

**BACH2**. *BACH2* encodes a transcription factor that plays critical roles in B and T cell biology, BACH2 deficient mice die of eosinophilic pneumonitis[28], and polymorphisms in the *BACH2* region have been associated with susceptibility to numerous immune-mediated diseases (Supplementary Table 10, Supplementary Data 1). The EGPA-associated variant, rs6454802, is in LD with variants associated with asthma, nasal polyps, and allergy. This genomic region is also associated with other immune-mediated diseases including celiac disease, IBD, PSC and multiple sclerosis (Supplementary Data 3).

**CDK6**. *CDK6* encodes a protein kinase that plays a role in cell cycle regulation. The EGPA-associated variant, rs42041, is associated with eosinophil count[13] and rheumatoid arthritis (Supplementary Table 10, Supplementary Data 3).

**10p14 intergenic region and GATA3**. The associated variant is not near any candidate genes, but CHiCP analysis shows interaction with the promoter of *GATA3* (Fig. 2d). *GATA3* encodes a master regulator transcription factor expressed by immune cells including T, NK, NKT, and ILC2 cells, that drives Th2 differentiation and secretion of IL-4, IL-5, and IL-13, and thus eosinophilic inflammation[29]. This genomic region is also associated with asthma and allergic rhinitis (Supplementary Table 11, Supplementary Data 1).

**HLA**. MPO+ EGPA was associated with a region encompassing the *HLA-DR* and *–DQ* loci (Supplementary Fig. 4). The classical *HLA* alleles at 2 or 4 digit resolution, and amino acid variants at 8 *HLA* loci, were then imputed. Using LMM, 9 *HLA* alleles conferring either susceptibility to or protection from MPO+ EGPA were identified (Supplementary Table 11). Conditional analyses revealed 3 signals conferred by 2 extended haplotypes encoding either *HLA-DRB1*0801-HLA-DQA1*04:01-HLA-DQB1*04:02*; or *HLA-DRB1*07:01-HLA-DQA1*02:01- HLA-DQB1*02:02/HLA-DQ*

*B1*03:03*; together with an additional signal at *HLA-DRB1*01:03* (Supplementary Table 11 and Supplementary Fig. 4). The strongest independent associations with disease risk were seen at *HLA-DRB1*08:01* (OR 35.8, $p = 7.6 \times 10^{-24}$), *HLA-DQA1*02:01* (OR 4.8, $p = 1.8 \times 10^{-15}$) and *HLA-DRB1*01:03* (OR 14.0, $p = 4.2 \times 10^{-8}$) (all *P*-values from LMM). Protection from disease was associated with the presence of *HLA-DQA1*05:01* (OR 0.4, $p = 1.2 \times 10^{-8}$). A similar analysis of the MHC signal seen in the ANCA-negative EGPA subset revealed no association with any of the imputed classical alleles. Analysis of *HLA* allelic frequencies stratified by country of recruitment revealed a consistent pattern (Supplementary Table 12), indicating that our findings were not the result of residual population stratification.

Individual amino acid variants in *HLA-DRB1*, *HLA-DQA1* and *HLA-DQB1* were associated with MPO+ ANCA EGPA (Supplementary Fig. 12A). Conditioning on the most associated amino acid variants at each locus, position 74 in HLA-DRB1, position 175 in HLA-DQA1 and position 56 in HLA-DQB1 demonstrated that HLA-DRB1 and HLA-DQB1 were independently associated with disease risk (Supplementary Fig. 12B-D). Conditioning on all three variants accounted for the entire signal seen at the MHC locus. The *HLA-DQ* locus associated with MPO+ EGPA appears the same as that previously associated with MPO+ ANCA-associated vasculitis[15]. To quantify the relative contribution of the MHC to the heritable phenotypic variance we partitioned the variance using BOLT-REML[30]. Using this approach, 6% of the heritable variance could be attributed to the MHC, an observation similar to that seen in other autoimmune diseases where the MHC contribution ranges from 2% in systemic lupus erythematosus to 30% in type 1 diabetes[31].

**Mendelian randomisation implicates eosinophils in aetiology**. The observation that eosinophilia is a ubiquitous clinical feature in EGPA does not necessarily imply that eosinophils play a causal role in the disease, since eosinophilia might instead be either an epiphenomenon or a downstream consequence of EGPA. The observation that 7 of the 8 alleles associated with increased EGPA risk are also associated with increased physiological eosinophil count in population-based cohorts, however, supports the notion of a causal role for eosinophils in EGPA. To formally test this hypothesis, we employed the technique of Mendelian randomisation (MR) (see the 'Methods' section). In contrast to observational associations, which are liable to confounding and/or reverse causation, MR analysis is akin to a natural randomised trial, exploiting the random allocation of alleles at meiosis to allow causal inference. MR analysis provided strong support ($P < 7.7 \times 10^{-12}$, inverse-variance weighted method) for a causal effect of eosinophil count on EGPA risk (Fig. 4b). This result was robust to a number of sensitivity analyses (see the 'Methods' section, Supplementary Fig. 13, Supplementary Data 4).

## Discussion

This study has identified 11 loci associated with EGPA, and reveals genetic and clinical differences between the MPO+ EGPA subset, and the larger ANCA-negative subset (with PR3+ ANCA patients too rare to be informative). There was a strong association of the MPO+ subset with *HLA-DQ*, and no *HLA-DQ* association with ANCA-negative EGPA. The ANCA-negative group alone was associated with variants at the *GPA33* and *IL5/IRF1* loci. There was clear evidence of association of both EGPA subgroups with variants at the *TSLP, BCL2L11* and *CDK6* loci, and suggestive evidence for *BACH2*, Chromosome 10, and *LPP*. A number of these loci have previously been shown to be associated with other autoimmune diseases, including PSC (Supplementary Table 10). Thus EGPA is characterised by certain genetic variants

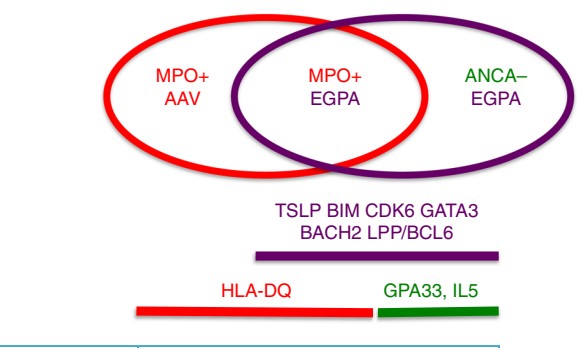

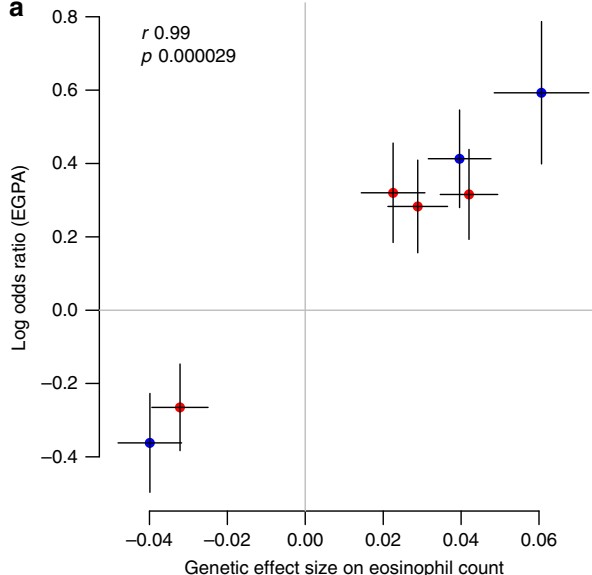

| | % of patients with feature | | |
|---|---|---|---|
| Clinical feature | MPO+ AAV (non EGPA) | MPO+ EGPA | ANCA– EGPA |
| Glomerulonephritis | 85 | 29* | 9 |
| Neuropathy | 20 | 79* | 57 |
| Asthma | n.d. | 100 | 100 |
| Eosinophilia | 4.5 | 100 | 100 |
| Pulmonary hemorrhage | 17 | 4 | 4 |
| Ear nose or throat | 32 | 81 | 88 |
| Pulmonary infiltrates | 20 | 45 | 61* |
| Cardiac involvement | 3 | 15 | 30* |
| Rituximab response | 98 | 80 | 38 |

**Fig. 3** Clinically and genetically distinct subsets within EGPA, and their relation to MPO+ AAV. Above: schematic showing relationship between MPO+ AAV, MPO+ EGPA and ANCA-negative EGPA, and putative genes underlying this classification. Below: Unshaded cells in the table show a comparison of the clinical features of MPO+ and ANCA-negative EGPA from this study as % (*$p < 0.0002$ compared to other EGPA subset: see Table 1), but also see refs. [4, 5]. Shaded cells show data from external sources: MPO+ AAV clinical data was derived from the EVGC AAV GWAS[15], and rituximab response rates for MPO+ AAV from the RAVE study[53] and for EGPA from ref. [32]. n.d. not determined

that associate with the syndrome as a whole, but others that indicate a genetic distinction between the MPO+ and ANCA-negative subsets. This distinction suggests important heterogeneity in the pathogenesis of the clinical syndrome, as the subsets have distinct clinical phenotypes and outcomes to therapy (Fig. 3). The increased efficacy of rituximab in the MPO+ subset of EGPA[32] suggests that future trials of novel therapies might need to be stratified according to ANCA status.

EGPA shares susceptibility variants with asthma (Table 2, Supplementary Table 4), and this genetic relatedness (Supplementary Fig. 2) allowed the use of the cFDR technique to detect additional EGPA-associated variants. For the *TSLP* variant, the estimated effect size in EGPA was much greater than in asthma (Supplementary Fig. 10), suggesting its association with asthma might be driven by a subset of patients. Consistent with this, rs1837253 (near *TSLP*) shows a trend towards association with a subset of severe asthmatics[12], as does the EGPA-associated variant at *C5orf56/IL5*. This raises the possibility that common EGPA-associated variants may be particularly associated with severe, adult-onset asthma– the asthma endotype typical of the EGPA prodrome. All or some of this subset of asthma patients may be at higher risk of subsequently developing EGPA.

Notably, 7 of 8 variants associated with EGPA (as a whole) were also associated with eosinophil count at genome-wide significance in normal individuals. The increase in EGPA risk conferred by each of the 7 genetic variants was proportional to its effect on eosinophil count (Fig. 4a), a correlation that held true when all eosinophil-associated variants were assessed for

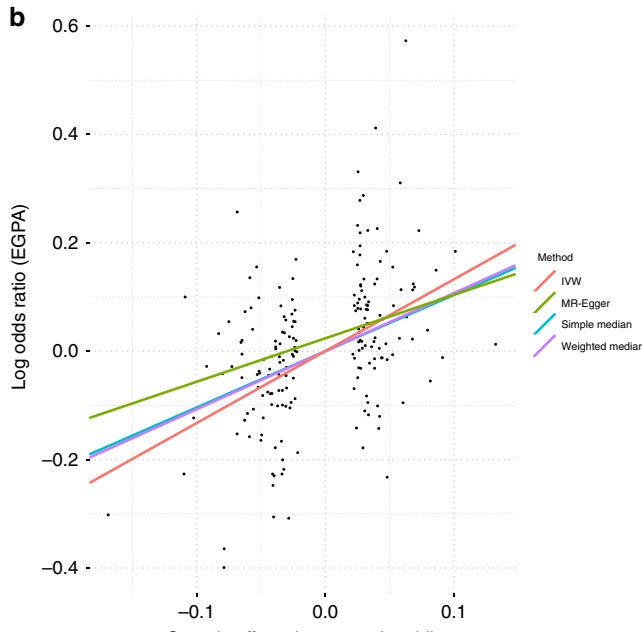

**Fig. 4** Relationship between genetic control of eosinophil count and risk of EGPA. **a** Correlation between the effect on EGPA risk and eosinophil count for the lead EGPA-associated genetic variants outside the *HLA* region. Blue points indicate variants discovered through standard case-control analysis, red points indicate variants discovered through cFDR. Horizontal and vertical lines indicate 95% confidence intervals. **b** Mendelian randomisation analysis supports a causal role for eosinophil abundance in EGPA aetiology. Points represent genome-wide significant conditionally independent variants associated with blood eosinophil count in the GWAS by Astle et al.[13] (where typed or reliably imputed in the EGPA dataset). Coloured lines represent estimated causal effect of eosinophil count on risk of EGPA from Mendelian randomisation (MR) methods. IVW inverse-variance weighted

EGPA risk (Supplementary Fig. 9). The causal nature of this association was supported by Mendelian randomisation analysis. This dose-response relationship between the genetic effects on eosinophil count and on risk of EGPA is consistent with a scenario where the genetic control of physiological variation in the eosinophil count contributes directly to the risk of pathological

eosinophilia and eosinophilic inflammatory disease. This inference is possible because the genetic effects on eosinophil count were derived in an independent dataset of healthy individuals (and thus the eosinophil count in this cohort is not confounded by the presence of disease). This concept that genetic variants that control a cell number within the normal range might, when combined, predispose to and perhaps initiate disease might extend beyond eosinophil-driven conditions. By analogy to studies establishing the relationship between genetic associations with LDL-cholesterol and with coronary artery disease risk[33], our data supporting a causal role for eosinophils could pave the way for novel eosinophil-targeted therapies in EGPA.

The shared eosinophil count and asthma-associated variants may explain the shared clinical features of the two EGPA subsets–both in the prodromal phase and after the development of frank EGPA. Moreover, they raise the possibility that persistent eosinophilia driven by variation in genes controlling physiological eosinophil levels can predispose to, or even directly give rise to, adult-onset asthma and to the EGPA prodrome (or to a progression from adult-onset asthma to the eosinophilic prodrome). After some time, an unknown proportion of people with this prodrome proceed to develop the clinical features characteristic of EGPA. Whether this progression is stochastic, is related to the severity of ongoing eosinophilic inflammation, is influenced by genetic factors or is triggered by an environmental factor remains unknown.

Around a third of EGPA patients develop ANCA against MPO and clinical features that overlap with MPO+ AAV, such as vasculitis and necrotising glomerulonephritis (Fig. 3). These patients appear to have a classic *HLA* class II-associated autoimmune disease with prominent eosinophilic features. The remaining patients, in contrast, develop disease characterised more by tissue eosinophilia, and its cardiac and pulmonary manifestations, in the absence of both autoantibodies or an association with a *HLA* class I or class II allele. This subgroup has a genetic association with expression of the barrier protein *GPA33*[24] and shared genetic architecture with IBD, suggesting that ANCA-negative EGPA might arise from mucosal/barrier dysfunction, rather than autoimmune disease.

EGPA has traditionally been treated with similar non-selective immunosuppression to AAV, such as steroids, cyclophosphamide or rituximab[2,32]. This study suggests investigation of further therapies may be warranted. Anti-IL5 (mepolizumab) has been developed for severe asthma, and efficacy in EGPA was recently confirmed in the MIRRA study[34]. Given the genetic variant at *IRF1/IL5* is associated with ANCA-negative EGPA, it would be interesting to specifically analyse this subset. Anti-TSLP agents are undergoing trials in asthma and might be considered in EGPA. BCL2 antagonists (e.g., ABT737) act by disrupting the sequestration of BIM by BCL2 and are being developed as cancer chemotherapeutics[35]. Their immunomodulatory effect[36] and exquisite sensitivity for driving mast cell apoptosis[37] suggests they may be effective in EPGA at low, perhaps non-toxic, doses. CDK6 inhibitors are also under development, and their impact in preclinical models[38] makes them additional therapeutic candidates for EGPA.

Our study had potential limitations. Whilst the MIRRA criteria are the most suitable diagnostic criteria for this study, it is possible that some patients with idiopathic hypereosinophilic syndrome may also meet these criteria. The choice of diagnostic criteria is discussed in detail in the Supplementary Note 1. We observed elevation of the genomic inflation factor even after adjustment for genetic principal components, suggesting potential residual population stratification. We therefore used a linear mixed model analysis which resulted in improved genomic control.

The rarity of EGPA makes GWAS challenging. Compared to GWAS of common diseases, our sample sizes were necessarily small and consequently our power was limited, particularly for uncommon or rare variants. Nevertheless, the majority of variants that achieved genome-wide significance in our primary cohort achieved nominal significance in our small replication cohort. Moreover, we observed strong correlation between estimated odds ratios at each EGPA-associated variant in our primary and replication cohorts. In addition, we were able to identify strong functional data and experimental literature to support the associations at the majority of loci identified (Supplementary Table 10). Finally, we highlight that the variant on chromosome 12 associated specifically with the MPO+ subgroup has a low minor allele frequency and was imputed rather than directly genotyped, and so its reproducibility will need to be evaluated in future larger studies.

In summary, this GWAS has demonstrated that EGPA is a polygenic disease. Most genetic variants associated with EGPA are also associated with control of the normal eosinophil count in the general population, suggesting a primary tendency to eosinophilia underlies susceptibility. Given the rarity of EGPA, it is likely that additional as yet unidentified environment or genetic factors are necessary to trigger disease. After the asthma/eosinophilia prodrome, EGPA develops and comprises two genetically and clinically distinguishable syndromes with different treatment responsiveness. MPO+ EGPA is an eosinophilic autoimmune disease sharing both clinical features and an MHC association with anti-MPO AAV. ANCA-negative EGPA may instead have a mucosal/barrier origin. Thus the identification of genes associated with EGPA helps explain its pathogenesis, points to logical therapeutic strategies, and supports a case for formally recognising the two distinct conditions that comprise it.

## Methods

**Inclusion criteria**. The important issue of diagnostic criteria is discussed in the Supplementary Note 1. We used the recently developed diagnostic criteria used in the Phase III clinical trial "Study to Investigate Mepolizumab in the Treatment of Eosinophilic Granulomatosis With Polyangiitis" (MIRRA: Supplementary Table 1)[34]. These define EGPA diagnosis based on the history or presence of both asthma and eosinophilia (>1.0 × 10$^9$/L and/or >10% of leucocytes) plus at least two additional features of EGPA.

**Subjects**. We recruited 599 individuals with a clinical diagnosis of EGPA from 17 centres in 9 European countries (Supplementary Table 2). Nine individuals were excluded because they did not fulfil the MIRRA criteria. In total, 534 patients were included in the GWAS after poor quality and duplicated samples and individuals with non-European ancestry were excluded (see section 'Genotyping and quality control' below). Of these, 294 (55%) were female and 240 (45%) male. Clinical characteristics are shown in Table 1. Three hundred and fifty two patients were ANCA-negative, 159 patients were MPO-ANCA+ve, and 5 were PR3-ANCA+ve. For 5 patients, there was no data on ANCA status. Thirteen patients were ANCA positive with either no data on specific antibodies to PR3 or MPO, or with positive ANCA immunofluorescence without detectable antibodies to PR3 or MPO.

Genotype data for 6000 UK controls was obtained from the European Prospective Investigation of Cancer (EPIC) Consortium. Four hundred and ninety eight individuals with a history of asthma were excluded. After QC, 5465 individuals remained. In addition, we recruited and genotyped controls from 6 European countries (Supplementary Table 2).

A further 150 patients with a clinical diagnosis of EGPA that fulfilled the MIRRA criteria were recruited from Germany and Italy for replication purposes, along with 125 controls from these countries. In total, 142 cases were included in the study following the removal of poor quality samples, of these 43 were MPO ANCA+ ve. All individuals provided written informed consent.

**Genotyping and quality control (QC)**. Genomic DNA was extracted from whole blood using magnetic bead technologies at the Centre for Integrated Genomic Medical Research (Manchester, UK) according to manufacturer's instructions. Patients with EGPA and healthy controls were genotyped using the Affymetrix UK Biobank Axiom array according to the manufacturer's protocol. Genotyping of cases and non-UK controls was performed by AROS Applied Biotechnology (Aarhus, Denmark). Genotyping of UK controls had been performed previously by

the EPIC-Norfolk consortium[39], also using the Affymetrix Axiom UK Biobank array.

Genotype calling was performed using the Affymetrix Powertools software. Calling was performed in batches of contemporaneously run plates as per the manufacturer's advice. After genotype calling, genetic data was processed in the following sequence using PLINK v1.9[40]. Samples with a sex mismatch, abnormal heterozygosity, or proportion of missing variants > 5%, were removed. Variants with missing calls >2%, deviation from Hardy-Weinberg Equilibrium ($p$-value < $1 \times 10^{-6}$), or which were monomorphic were removed. The QC process was performed separately for each batch. The post-QC genotype data from each batch was then merged. Following this merger, duplicated or related samples (identified using Identity by State) were removed, and the genetic variant QC was performed again so that variants with missing calls >2% or deviation from Hardy-Weinberg Equilibrium ($p$-value < $1 \times 10^{-6}$) in the combined data were excluded. Variants that showed significant differential missingness (Benjamini-Hochberg adjusted $p$-value < 0.05) between cases and controls were removed. Finally, principal components analysis (PCA) of the post-QC genotype calls combined with calls from 1000 Genome individuals was performed (Supplementary Fig. 14). Samples of non-European ancestry by PCA were excluded as described below. The means and standard deviations of PC1 and PC2 were calculated for the EUR subset of the 1000 Genomes samples. Cases or controls lying outside $+/-$ 3 standard deviations from the mean on either PC1 or PC2 were removed. Following these QC steps, 534 patients and 6688 controls remained (see Supplementary Table 2). 543,639 autosomal variants passed QC.

**Replication cohort genotyping.** Replication cohort samples were genotyped using the Affymetrix UK Biobank Axiom array according to the manufacturer's protocol by Cambridge Genomic Services (Cambridge, UK). Genotype calling and QC was performed as outlined above except that samples were processed as a single batch. Following these steps, 142 cases, 121 controls and 626,229 autosomal genetic variants passed QC (Supplementary Table 5).

**Association testing on the directly genotyped data.** Case-control association testing on the 543,639 directly genotyped autosomal variants was initially performed using logistic regression with the SNPTEST software. Principal components (PCs) were included as covariates to adjust for confounding factors. We calculated PCs on the genotype matrix containing both cases and controls. Whilst this carries a small risk of masking true disease associations, we felt that this more conservative approach was appropriate since (i) the UK controls were genotyped in a different facility thereby potentially leading to confounding from a batch effect, and (ii) we did not have French controls. QQ plots of expected test statistics under the null hypothesis of no genotype-disease association vs. the observed test statistics are shown in Supplementary Fig. 15. The genomic inflation factor (lambda, the ratio of the median of the observed chi-squared statistics to the median expected chi-squared statistic under the null hypothesis) was calculated. We assessed the effect of using increasing numbers of PCs as covariates on lambda. Using the first 20 PCs produced a reduction in lambda to 1.09, with little benefit from the inclusion of further PCs.

**Pre-phasing and imputation.** Imputation was performed using the 1000 genomes phase 3 individuals as a reference panel (data download date 7th July 2015). Pre-phasing was first performed with SHAPEIT[41], and then imputation was performed using IMPUTE2[42,43]. For the IMPUTE2 Monte Carlo Markov Chain algorithm we used the default settings (30 iterations, with the first 10 discarded as the 'burn-in' period). The 'k_hap' parameter was set to 500. IMPUTE2 was provided with all available reference haplotypes from the 1000 Genomes individuals, as the software chooses a custom reference panel for each individual to be imputed.

**Association testing following imputation.** Association testing on the imputed data was initially carried out using the SNPTEST software. We used an additive genetic model (option '--frequentist 1'). Uncertainty in the imputed genotypes was taken into account in the association testing by using a missing data likelihood score test. Despite inclusion of the first 20 PCs as covariates, the genomic inflation factor for the association tests using the imputed data was 1.10.

Since lambda was 1.10 despite inclusion of 20 PCs as covariates, suggesting residual population stratification, we performed the GWAS using a linear mixed model with the BOLT-LMM software[44]. 9,246,221 autosomal variants either directly typed or imputed with an info metric greater than 0.7 were taken forward for association testing. This approach resulted in an improved lambda value of 1.047.

**Analysis stratified by ANCA.** In addition to testing all EGPA cases vs. controls, we also tested MPO+ cases ($n = 159$) against controls, and ANCA-negative cases ($n = 352$) against controls. These subset analyses were performed using BOLT-LMM. Five individuals who were PR3-ANCA positive and 18 individuals in whom we had either no data on ANCA status, or who were ANCA positive by immunofluorescence but in whom specific antibodies to MPO and PR3 were negative or

unknown were excluded from these subset analyses. In addition, we performed a within-cases genetic analysis, comparing MPO+ cases ($n = 159$) against ANCA-negative cases ($n = 352$). This analysis was performed with logistic regression with inclusion of 10 principal components. Lambda for this analysis was 1.037.

**Leveraging association statistics from related traits.** In order to increase power to detect EGPA-associated genetic variants, we used a conditional FDR (cFDR) method[10,11] to leverage findings from other GWAS studies of related phenotypes. We used the summary statistics from a GWAS of asthma[12] and a population-scale GWAS of peripheral blood eosinophil counts[13] to calculate a conditional FDR for each variant for association with EGPA conditional on each of these other traits. We used the P-values for EGPA from the linear mixed model analysis with BOLT-LMM.

Given p-values $P_i$ for EGPA and $P_j$ for the conditional trait (eg asthma), and a null hypothesis of no association with EGPA $H_0$, the cFDR for p-value thresholds $p_i$ and $p_j$ is defined as

$$cFDR(p_i, p_j) = \Pr(H_0 | P_i \leq p_i, P_j \leq p_j)$$

$$\leq p_i \frac{\#(\text{variants with } P_i \leq p_i)}{\#(\text{variants with } P_i \leq p_i, P_j \leq p_j)}$$

roughly analogous to Storey's $q$-value computed only on the variants for which $P_j \leq p_j$.

The cFDR has the advantage of asymmetry, only testing against one phenotype at a time. Intuitively, if we know that associations are frequently shared between asthma and EGPA, and attention is restricted to a set of variants with some degree of association with asthma, we may relax our threshold for association with EGPA. The cFDR formalises this intuition in a natural way, with the adjustment in the threshold responding to the total degree of observed overlap between disease associations.

The cFDR lacks a convenient property of the Q-value limiting the overall FDR to the significance threshold for the test statistic. Namely, if we reject $H_0$ at all variants with cFDR < $\alpha$, the overall FDR is not necessarily bound above by $\alpha$[11]. An upper bound on the overall FDR can be obtained by considering the region of the unit square for which $p_i$, $p_j$ reaches significance, with the bound typically larger than $\alpha$.

We used a threshold $\alpha = $ cFDR ($5 \times 10^{-8}$, 1), chosen to be the most stringent threshold for which all variants reaching genome-wide significance in a univariate analysis will also reach significance in the cFDR analysis. The value $\alpha$ is equal to the FDR for this univariate analysis on the set of variants for which the cFDR is defined. Put more intuitively, cFDR thresholds for the analyses conditioning on asthma and eosinophilia were chosen so that the overall FDR was less than the FDR corresponding to a P value for genome-wide significance in the standard univariate analysis.

Clearly, the cFDR method requires that genotype data for a given variant is available for both traits. The cFDR analysis was limited to variants that were directly typed in each GWAS. The cFDR analysis for EGPA conditional on asthma examined 74,776 variants and that for EGPA conditional on eosinophil count 513,801.

**Replication cohort analysis.** Significant variants from the primary cohort analysis (identified either through the linear mixed model analysis or the cFDR method) were tested in the replication cohort using SNPTEST and including 2 principal components as covariates to control for population stratification. The results of the two cohorts were meta-analysed using the META software using an inverse-variance method based on a fixed-effects model.

**Fine-mapping.** To identify the most likely causal variant we performed fine-mapping as follows. We computed approximate Bayes factors for each variant using the Wakefield approximation[45], with a prior parameter $W = 0.09$, indicating our prior expectation that true effect sizes (relative risks) exceed 2 only 1% of the time. Thus, we calculated posterior probabilities that each variant is causal, assuming a single causal variant per LD-defined region as previously proposed[46]. Code to perform these steps is available at https://github.com/chr1swallacw/lyons-et-al.

**HLA imputation.** Two thousand seven hundred and seventy seven SNPs, 343 classical HLA alleles to 2 or 4 digit resolution and 1438 amino acid variants were imputed at 8 HLA loci (HLA-A, HLA-B, HLA-C, HLA-DRB1, HLA-DQA1, HLA-DQB1, HLA-DPA1 and HLA-DPB1) from phased genotype data using the HLA*IMP:03 software[47]. Association testing for HLA variants was performed with BOLT-LMM.

**Narrow-sense heritability estimation.** The narrow-sense heritability ($h^2$) of EGPA was estimated using LD-score regression[48]. LD scores were calculated using the European 1000 Genomes Project reference panel. To estimate the contribution of the MHC to heritable phenotypic variance we performed a variance-components analysis using BOLT-REML[30].

**Promoter enhancer interaction mining.** Long-range interactions between genetic variants associated with EGPA and gene promoter and regulatory regions were identified in promoter capture Hi-C datasets from a range of primary cell types[49] and cell lines[50] using the CHiCP browser[51].

**Data mining.** Other traits associated with EGPA-associated loci were identified using PhenoScanner[52]. Phenoscanner searches GWAS data from multiple sources including the NHGRI-EBI (National Human Genome Research Institute-European Bioinformatics Institute) GWAS Catalogue and NHLBI GRASP (Genome-Wide Repository of Associations Between SNPs and Phenotype) Catalogue, and accounts for LD between queried genetic variants and those in the catalogues of trait-associated variants. For each locus associated with EGPA, we searched for traits associated with variants in LD ($r^2$ 0.6 or higher) with the lead EGPA variant. We searched for all associations with a p-value of $1 \times 10^{-5}$ or lower so that in addition to identifying all associations that have achieved genome-wide significance ($P < 5 \times 10^{-8}$), we also identified associations that are suggestive/sub-genome wide. We also used PhenoScanner to identify eQTLs at EGPA-associated loci. Run date 22nd July 2019.

**Associations of clinical characteristic with ANCA status.** Comparisons of clinical features between MPO+ EGPA patients ($n = 159$) and ANCA-negative patients ($n = 352$) were performed from $2 \times 2$ contingency tables using 1 degree of freedom chi-squared tests with Yates' continuity correction. Bonferroni correction of $P$ values was undertaken to account for the multiple testing of 8 clinical features. Associations with PR3 antibodies were not assessed as this subgroup ($n = 5$) was too small for statistical analysis. Patients with missing or incomplete ANCA data, and those who were ANCA positive by immunofluorescence without MPO or PR3 antibodies were excluded from this analysis.

ANCA status was tested for association with country of origin using a chi-squared test on the $8 \times 2$ contingency table (Supplementary Table 7). Columns represented ANCA status (ANCA-negative or MPO+) and rows represented country of origin. The Republic of Ireland and the UK were considered as one entity for the purposes of this analysis. We found a significant association between ANCA status and country of origin ($P$ $7.6 \times 10^{-10}$). Since there were only 5 cases from the Czech Republic, in sensitivity analysis, we repeated the chi-squared test after merging the counts for Czech and German cases. This did not materially affect the association ($P$ $1.9 \times 10^{-9}$). This association is likely to reflect the differing specialities of recruiting centres (ANCA positive patients are more likely to be found in nephrology clinics than rheumatology clinics). Nevertheless, to ensure that the clinical associations that we identified with ANCA status were not in fact driven by geographical differences, we repeated the association testing adjusting for country of origin. We did this by performing logistic regression of each clinical feature on ANCA status (MPO+ vs. ANCA-negative), with country of origin (coded using dummy variables) as a covariate (Supplementary Table 8).

**Mendelian randomisation.** "Two-sample" Mendelian randomisation (MR) was performed to assess whether there is a causal effect of eosinophil count (the exposure) on EGPA (the outcome). MR analysis was performed using the MendelianRandomization R package. Summary statistics were obtained for 209 conditionally independent variants associated with peripheral blood eosinophil count in a population study by Astle et al.[13]. One hundred and ninety three of these were typed or imputed in the EGPA dataset. MR analysis was performed using the beta coefficients and standard errors for these 193 variants in the eosinophil count GWAS and in our EGPA GWAS (all EGPA vs. controls using BOLT-LMM). The beta coefficients from BOLT-LMM are distinct from the log(OR) obtained from logistic regression. Therefore beta coefficients from BOLT-LMM and their standard errors were transformed by dividing them by ($\mu \times (1 - \mu)$), where $\mu$ is the case fraction, to provide estimates comparable to traditional log(ORs). These transformed betas and standard errors were then used in the MR analysis. The primary MR analysis was conducted using the inverse-variance weighted method. Additional sensitivity analyses were performed using alternative methods less susceptible to violation of MR assumptions by pleiotropy (Supplementary Fig. 13).

**Study approval.** Written informed consent was received from participants prior to inclusion in the study. Details of ethical approval for each participating centre are shown in Supplementary Table 3.

**Reporting summary.** Further information on research design is available in the Nature Research Reporting Summary linked to this paper.

## Data availability
The authors declare that all data supporting the findings of this study are available within the paper and its supplementary information files. Per-variant genetic association summary statistics are included as Source Data File.

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

## Acknowledgements

This work was funded primarily by Project Grants from Arthritis Research UK (20593 to Drs. Smith and Lyons) and the British Heart Foundation (PG/13/64/30435 to Drs. Smith and Lyons). Additional support was provided by the NIHR Cambridge Biomedical Research Centre, the West Anglia Comprehensive Research Network, a Medical Research Council Programme Grant (MR/L019027/1 to Dr. Smith), a Wellcome Trust Investigator Award (200871/Z/16/Z to Dr. Smith), a NIHR Senior Investigator Award (to Dr. Smith), a Wellcome Trust Senior Research Fellowship (WT107881 to Dr. Wallace), a Medical Research Council grant (MC_UU_00002/4 to Dr. Wallace), a Wellcome Trust Mathematical Genomics and Medicine Programme Studentship (to Dr. Liley), a Career Development Award from the Cambridge British Heart Foundation Centre for Research Excellence and a UK Research Innovation Fellowship (RE/13/6/30180 and MR/S004068/1 to Dr. Peters). Additional aspects of this work were supported by the following funding: a Science Foundation Ireland Grant (11/Y/B2093 to Dr Little); an Australian National Health and Medical Research Council (NH&MRC), Career Development Fellowship (ID 1053756) and a Victorian Life Sciences Computation Initiative (VLSCI) grant number (VR0240) on its Peak Computing Facility at the University of Melbourne, an initiative of the Victorian Government, Australia (to Dr Leslie); Research at the Murdoch Children's Research Institute was supported by the Victorian Government's Operational Infrastructure Support Program; Project RVO 64 165 of the Ministry of Health of Czech Republic (to Dr Tesar); Ministerio de Economía y Competitividad (SAF SAF 2017-88275-R), FEDER una manera de hacer Europa) and CERCA programme (to Drs Cid and Hernández-Rodríguez) and Instituto de Salud Carlos III (PI 18/00461) (to Drs Espígol-Frigolé & Prieto-Gonzalez); Prof Bruce is an NIHR Senior Investigator and is supported by Versus Arthritis, the National Institute for Health Research Manchester Biomedical Research Centre and the NIHR Manchester Clinical Research Facility. We thank the Centre for Integrated Genomic Medical Research Biobank for sample storage and preparation, and AROS Applied Biotechnology (Aarhus, Denmark) and Cambridge Genomic Services (Cambridge, UK) for genotyping.

## Author contributions

K.G.C.S. and P.A.L. conceived and designed the study. P.A.L., J.E.P., J.L., R.M.R.C., F.A., S.L., D.V., W.A., H.E., T.J. and C.W. analysed data. F.A., C.B., M.C.C., J.E., L.G., D.R.W.J., I.G., P.L., M.A.L., D.M., F.M., S.O., G.A.R., B.R., G.S., R.A.S., W.S., V.T., B.T., R.A.W., A.V., J.U.H. and the E.V.G.C. provided samples and collected clinical data. K.G.C.S. wrote the paper with P.A.L., J.E.P. and J.L. All authors read and approved the final version of the paper.

## Competing interests

S.L. is a partner in Peptide Groove LLP. The remaining authors declare no competing interests.

## Additional information

Paul A Lyons[1,2,65], James E Peters [1,3,4,65], Federico Alberici[1,5,6,65], James Liley[1,7,65], Richard M.R. Coulson[1], William Astle [3,7,8], Chiara Baldini[9], Francesco Bonatti[10], Maria C Cid[11], Heather Elding[12,13], Giacomo Emmi [14], Jörg Epplen[15], Loïc Guillevin[16], David R.W. Jayne[1], Tao Jiang[3], Iva Gunnarsson[17], Peter Lamprecht[18], Stephen Leslie[19,20], Mark A. Little[21], Davide Martorana[10], Frank Moosig[22], Thomas Neumann[23,24], Sophie Ohlsson [25], Stefanie Quickert[23,26], Giuseppe A. Ramirez[27], Barbara Rewerska[28], Georg Schett[29], Renato A. Sinico[30], Wojciech Szczeklik[28], Vladimir Tesar [31], Damjan Vukcevic [19,20], The European Vasculitis Genetics Consortium, Benjamin Terrier[16], Richard A Watts [32,33], Augusto Vaglio[34], Julia U Holle[22], Chris Wallace[1,2,7] & Kenneth G.C. Smith[1,2]*

[1]Department of Medicine, University of Cambridge School of Clinical Medicine, University of Cambridge, Cambridge Biomedical Campus, Cambridge CB2 0QQ, UK. [2]Cambridge Institute for Therapeutic Immunology and Infectious Disease, Jeffrey Cheah Biomedical Centre University of Cambridge, Cambridge CB2 0AW, UK. [3]BHF Cardiovascular Epidemiology Unit, Department of Public Health and Primary Care, University of Cambridge, Strangeways Research Laboratory, Wort's Causeway, Cambridge CB1 8RN, UK. [4]Health Data Research UK, Cambridge, UK. [5]Nephrology and Immunopathology Unit-ASST Santi Paolo e Carlo, San Carlo Borromeo Hospital, Milan, Italy. [6]Dipartimento di Scienze della Salute, University of Milano, Milano, Italy. [7]Medical Research Council Biostatistics Unit, Cambridge Institute of Public Health, Cambridge Biomedical Campus, Forvie Site, Robinson Way, Cambridge CB2 0SR, UK. [8]NHS Blood and Transplant, Long Road, Cambridge Biomedical Campus, Cambridge, UK. [9]Rheumatology Unit, University of Pisa, Pisa, Italy. [10]Unit of Molecular Genetics, University Hospital of Parma, Via Gramsci 14, 43126 Parma, Italy. [11]Department of Autoimmune Diseases, Hospital Clínic, University of Barcelona, Institut d'Investigacions Biomèdiques August Pi i Sunyer (IDIBAPS), CRB-CELLEX, Barcelona, Spain. [12]The National Institute for Health Research Blood and Transplant Unit in Donor Health and Genomics at the University of Cambridge, University of Cambridge, Strangeways Research Laboratory, Wort's Causeway, Cambridge CB1 8RN, UK. [13]Department of Human Genetics, The Wellcome Trust Sanger Institute, Wellcome Trust Genome Campus, Hinxton, Cambridge CB10 1HH, UK. [14]Department of Experimental and Clinical Medicine, University of Firenze, Firenze, Italy. [15]Human Genetics, Ruhr University Bochum, Bochum, Germany. [16]Service de Médecine Interne, Hôpital Cochin, 75679 Paris Cedex 14, France. [17]Division of Rheumatology, Department of Medicine, Karolinska University Hospital, Karolinska Institute, Stockholm, Sweden. [18]Department of Rheumatology and Clinical Immunology, University of Lübeck, 23562 Lübeck, Germany. [19]Schools of Mathematics and Statistics, and BioSciences, and Melbourne Integrative Genomics, University of Melbourne, Parkville, VIC 3010, Australia. [20]Data Science, Murdoch Children's Research Institute, Parkville, VIC 3052, Australia. [21]Trinity Health Kidney Centre, Trinity Translational Medicine Institute, Tallaght Hospital, Dublin, Ireland. [22]Rheumazentrum Schleswig-Holstein Mitte, Neumünster, Germany. [23]Department of Internal Medicine 3, Jena University Hospital, Jena, Germany. [24]Department of Rheumatology, Immunology and Rehabilitation, Cantonal Hospital St. Gallen, St. Gallen, Switzerland. [25]Department of Nephrology, Division of Clinical Sciences, Lund University, Lund, Sweden. [26]Department of Internal Medicine 4 (Gastroenterology, Hepatology, and Infectious Diseases), Jena University Hospital, Jena, Germany. [27]Unit of Immunology, Rheumatology, Allergy and Rare Diseases, Università Vita Salute San Raffaele and IRCCS Ospedale San Raffaele, Milan, Italy. [28]Jagiellonian University Medical College, Kraków, Poland. [29]Department of Internal Medicine 3, Rheumatology and Immunology, Friedrich Alexander University Erlangen-Nuremberg and Universitatsklinikum Erlangen, Erlangen, Germany. [30]Department of Medicine and Surgery, Università degli Studi di Milano–Bicocca (School of Medicine and Surgery), via Cadore, 48, 20900 Monza, Italy. [31]Department of Nephrology, 1st Faculty of Medicine and General University Hospital, Charles University, Prague, Czech Republic. [32]Department of Rheumatology, Ipswich Hospital, Heath Road, Ipswich, Suffolk IP4 5PD, UK. [33]Norwich Medical School, University of East Anglia, Norwich NR7 4TJ, UK. [34]Department of Biomedical Experimental and Clinical Sciences "Mario Serio", University of Firenze, and Meyer Children's Hospital, Firenze, Italy. [65]These authors contributed equally: Paul A Lyons, James E Peters, Federico Alberici, James Liley. A full list of consortium members and their affiliations appears at the end of the paper.

## The European Vasculitis Genetics Consortium

Mohammed Akil[35], Jonathan Barratt[36], Neil Basu[37], Adam S. Butterworth[3,4,12], Ian Bruce[38,39], Michael Clarkson[40], Niall Conlon[41], Bhaskar DasGupta[42], Timothy W.R. Doulton[43], Georgina Espígol-Frigolé[11], Oliver Flossmann[44], Armando Gabrielli[45], Jolanta Gasior[46], Gina Gregorini[47], Giuseppe Guida[48], José Hernández-Rodríguez[11], Zdenka Hruskova[31], Amy Hudson[20], Ann Knight[49], Peter Lanyon[50], Raashid Luqmani[51], Malgorzata Magliano[52], Angelo A. Manfredi[27], Christopher Marguerie[53], Federica Maritati[34], Chiara Marvisi[34], Neil J. McHugh[54], Eamonn Molloy[55], Allan Motyer[19], Chetan Mukhtyar[56], Leonid Padyukov[17], Alberto Pesci[57], Sergio Prieto-Gonzalez[11], Marc Ramentol-Sintas[58], Petra Reis[29], Dario Roccatello[59], Patrizia Rovere-Querini[26], Carlo Salvarani[60], Francesca Santarsia[61], Roser Solans-Laque[58], Nicole Soranzo[13,62], Jo Taylor[63], Julie Wessels[64] & Jochen Zwerina[29]

[35]Sheffield Royal Hallamshire Hospital, Sheffield S10 2JF, UK. [36]Department of Infection, Immunity and Inflammation, University of Leicester, Leicester LE1 9HN, UK. [37]Institute of Infection, Immunology and Inflammation, University of Glasgow, Glasgow, UK. [38]NIHR Manchester Musculoskeletal Biomedical Research Centre, Manchester University Hospitals NHS Foundation Trust, Manchester, UK. [39]Centre for Epidemiology Versus Arthritis, Centre for Musculoskeletal Research, The University of Manchester, Manchester Academic Health Science Centre, Manchester, UK. [40]Cork University Hospital, Cork, Ireland. [41]Department of Immunology, St James' Hospital Dublin, Dublin, Ireland. [42]Southend University

Hospital, Westcliff-on-Sea SS0 0RY, UK. [43]East Kent Hospitals University NHS Foundation Trust, Kent and Canterbury Hospital, Canterbury CT1 3NG, UK. [44]Royal Berkshire Hospital NHS Trust, Reading RG1 5AN, UK. [45]Department of Internal Medicine, Ospedali Riuniti-Università, Politecnica delle Marche, Ancona, Italy. [46]University Hospital, Department of allergy and immunology, Krakow, Poland. [47]Nephrology Unit, Spedali Civili, Brescia, Italy. [48]Internal Medicine II, Immunology and Allergology Outpatient Clinic, Medical Science Department, ASL TO2 Birago di Vische Hospital, and the University of Torino, Turin, Italy. [49]Department of Medical Sciences, Uppsala University, 751 85 Uppsala, Sweden. [50]Nottingham University Hospitals NHS Trust, Nottingham NG7 2UH, UK. [51]Nuffield Orthopaedic Centre, Oxford OX3 7LD, UK. [52]Stoke Mandeville Hospital, Aylesbury HP21 8AL, UK. [53]South Warwickshire NHS Foundation Trust, Warwick, UK. [54]Royal National Hospital for Rheumatic Disease, Bath BA1 1RL, UK. [55]St Vincent's Hospital Dublin, Dublin, Ireland. [56]Norfolk and Norwich University Hospital, Norwich NR4 7UY, UK. [57]Pneumology Unit, University of Milano Bicocca, Milan, Italy. [58]Research Unit in Systemic Autoimmune Diseases, Vall d'Hebron Research Institute, Hospital Vall d'Hebron, Barcelona, Spain. [59]Department of Rare, Immunologic, Hematologic and Immunohematologic Diseases, Center of Research of Immunopathology and Rare Diseases, University of Torino, Torino, Italy. [60]Azienda USL-IRCCS di Reggio Emilia e Università di Modena e Reggio Emilia, Reggio Emilia, Italy. [61]Nephrology Unit, University Hospital of Parma, Parma, Italy. [62]Department of Haematology, University of Cambridge, Cambridge Biomedical Campus, Cambridge CB2 0PT, UK. [63]Dorset County Hospital, Dorchester DT1 2JY, UK. [64]Royal Stoke University Hospital, Stoke-on-Trent ST4 6QG, UK

