## [Peer Review File · Nature Communications]

Reviewers' Comments:

Reviewer #1:

Remarks to the Author:

This is a novel and interesting paper that provides potentially important, potential insights into the pathogenesis of EGPA. The paper is generally well written and data moderately compelling. I have several questions and comments as below:

Major Points

1. Genetic inflation factor correction assumes that the level of inflation is uniform across the genome, when we know it is not. The lambda of 1.09 reported at line 558, was this lambda(1000) or raw, and was it including or excluding the MHC? Have the authors tried alternate methods to control stratification such as a. dividing into strata and then meta-analysing, or b. using linear mixed models instead of PCA?
2. The cross-phenotype analysis and fine-mapping has not been performed in depth and some usual methods in this form of study have not been performed, and supportive data from other diseases not mentioned (see also minor point 4 and 5). Association of the chromosome 2q locus harbouring BCL2L11 has also been reported with multiple cell types from blood counts, and IgA nephropathy, and CLL, as well as the IBD, PSC and eosinophil counts that are reported. These should be mentioned – as it stands the association reporting is selective. Are the associations concordant? How are the investigators sure that this locus rather than ACOXL1 is EGPA associated? Have the investigators done any of the Mendelian randomization methods such as SMR to try to pick the key gene? The same goes for the other reported loci.
3. How was the HLA association testing 'adjusted for population structure' when the main analysis suggested that with 20PCs structure effects could not be controlled (line 666-667)? In the methods it is stated that subjects of non-European ancestry were excluded. Typically the PCA is used to define ancestry and those beyond a particular limit (usually 6SDs on PC1 or PC2) are excluded. What was done here? Method should be mentioned including the PC cutoffs (so that the excluded regions can be assessed in Supplementary Figure 10). Is Supplementary 11 pre or post stratification control?
4. The HLA analysis presented in Supplementary Figure 8B suggests that HLA-DRB1*0103 is the main-EGPA associated variant but the text lists this as the 3rd locus. Supplementary Table 12 indicates this is incorrect. This allele is IBD and AS-associated, both diseases that are vasculitis associated. There's no mention of this in the Discussion. Why not – this is a massive effect so it's a particularly strange omission? Also, for the key HLA alleles that are associated, the ORs should be mentioned in the main text?
5. The authors make the suggestion that ANCA-negative disease may have a gut/mucosal origin rather than being autoimmune. Are they suggesting that the disease is autoreactive, or what pathological process are they suggesting? The data to support this as presented is pretty lean (although the authors have not yet apparently factored in the genetic overlap with known gut mucosal diseases such as IBD and AS). A simple test that should be done is to look at the overlap between the genetic associations and epigenetic marks in different cell and tissue types, such as using the program EPIGWAS. This could provide statistical support for what otherwise looks like a hypothesis only.

Minor Points

1. What is the contribution of the MHC vs the non-MHC loci in EGPA?
2. What is the genetic variance between ANCA-positive and –negative EGPA, and if the datasets are available, with asthma?
3. Regarding replication, what's the correlation of log(OR) for EGPA vs the replication dataset? High correlation supports replication (as per Figure 3a)?
6. How does the data indicate that EGPA is polygenic rather than oligogenic?
7. BACH2 is also associated with PSC and AS 1, both diseases associated with vasculitis. Neither is mentioned. As per major point 2, is the association concordant or discordant?
8. SOCS1 is associated with AS, PSC and IBD 1. This is not mentioned; all three diseases are associated with vasculitis, and IBD/PSC with ANCA. As per point 4, are the associations concordant

or discordant? This information may help demonstrate that this locus is a true positive. Surely this is more relevant than ChIP data from NOD mice??

9. Please provide the zoom plots of the reported loci in the Supplement.

Reference:

1. Ellinghaus D, et al. Analysis of five chronic inflammatory diseases identifies 27 new associations and highlights disease-specific patterns at shared loci. Nat Genet 48, 510-518 (2016).

Reviewer #2:

Remarks to the Author:

The authors demonstrate that EGPA is polygenic with genetic distinctions between MPO+ and ANCA negative disease, correlating with different clinical features.

This is well written, the tables and figures are of high quality, and discussion was clearly presented.

Reviewer #3:

Remarks to the Author:

This paper reads very well – no major objections in terms technical flaws etc.

The figures 1 lack an x-axis and what happened to the sex chromosomes? I was wondering if there isn't a prettier and more informative way to integrate these results into a single plot -- perhaps using different colors for the subset specific loci, and the key loci annotated? (In this way it feels like Figures 1 and 2 are very different – made by different people?) It also seems crazy in 2018 not to present the full genome-wide analysis results (including chrX). On a separate note, I strongly feel the publication of the manuscript must be accompanied with public release of the summary stats of the major strata described here. There needs to be stricter enforcement from the journal.

For figure 3 B I did not think the legend was sufficiently clear - I didn't really understand what was plotted (the numbers given). I would have found it more intuitive if the authors presented genetic correlation estimates (genome-wide) between the GPA subsets and the traits listed.

Table 1. The table header seems more like a conclusion than a factual description of the table contents. Is there more clinical information wrt the (sub)type of cardiomyopathy observed in these patients? Spell out ENT.

The most interesting finding is the potential role of eosinophil levels – I would have loved to see an MR analysis done to test whether such levels are in fact causally driving EGPA or asthma etc?

We would like to thank the reviewers for their insightful comments and suggestions regarding our manuscript. We are pleased they found our work novel and interesting, and providing insight into the pathogenesis of EGPA. We have addressed their specific points, including adding new analyses where appropriate, as detailed below. Reviewer comments are in *italics*, and additions to the text are quoted in purple.

Reviewer #1 (Remarks to the Author):

This is a novel and interesting paper that provides potentially important, potential insights into the pathogenesis of EGPA. The paper is generally well written and data moderately compelling. I have several questions and comments as below:

Major Points

1. Genetic inflation factor correction assumes that the level of inflation is uniform across the genome, when we know it is not. The lambda of 1.09 reported at line 558, was this lambda(1000) or raw, and was it including or excluding the MHC? Have the authors tried alternate methods to control stratification such as a. dividing into strata and then meta-analysing, or b. using linear mixed models instead of PCA?

The lambda we report is lambda raw and excludes the MHC. Given the pan-European nature of our primary cohort, we included the first 20 principal components as covariates in the logistic regression model to address the potential issue of population stratification. While this controls inflation to some extent it is clearly not perfect, as reflected by the lambda value of 1.09 on the typed data and 1.10 on the imputed data. We agree that the genomic inflation correction assumes uniform levels of inflation, and we now acknowledge this is in the description of the genomic control procedure in the Methods (lines 671-673).

“A potential limitation of this genomic inflation correction procedure is that it assumes the level of inflation is uniform across the genome.”

Following the helpful suggestion of the reviewer, we have now re-run our analyses using the linear mixed model approach implemented in BOLT-LMM (Loh et al Nature Genetics, 2015). This approach leads to a significantly reduced lambda (1.047 for all EGPA versus controls), and the associations we observed with EGPA as a whole and in the ANCA negative and MPO positive subsets are preserved. The results of these new analyses are presented in the new Supplementary Table 4.

We have modified the text of the manuscript to reflect this new data as follows (lines 140-156).

“Despite attempting to control for population stratification by the inclusion of 20 genetic principal components (PCs) as covariates in the logistic regression model, the genomic inflation factor lambda remained 1.10. To account for residual genomic inflation, we calculated an adjusted genome-wide significance p-value threshold of 1×10^{-8} (**Methods**). Use of this adjusted threshold is equivalent to using a p-value threshold of 5×10^{-8} if there were no elevation of the genomic inflation factor (assuming that the level of inflation is uniform across the genome). Three genetic associations met this adjusted threshold (**Figure 1A, Supplementary Figure 1, Table 2**). The strongest association was with the MHC, and the others were on chromosome 2 near *BCL2L11* (encoding Bim), and on chromosome 5 near the *TSLP* gene (which encodes Thymic stromal lymphopoietin; TSLP). Since adjustment for genetic PCs did not completely abrogate genomic inflation, we used a linear mixed model as an alternative method to control for population stratification (**Methods**). This method more effectively controlled genomic inflation (lambda 1.047) and the 3 associations were preserved at genome-wide significance

(Supplementary Table 4), providing strong evidence against these signals being driven by population stratification.”

In addition, we have added the following text (lines 188-189)

“These 7 additional associations were preserved in the mixed model analysis”

We have altered the Discussion to say (lines 516-519)

“...and so we used a more stringent significance threshold adjusted for residual genomic inflation to mitigate against this. In addition the use of a linear mixed model analysis improved the value of lambda and confirmed the reported associations.”

2. The cross-phenotype analysis and fine-mapping has not been performed in depth and some usual methods in this form of study have not been performed, and supportive data from other diseases not mentioned (see also minor point 4 and 5). Association of the chromosome 2q locus harbouring BCL2L11 has also been reported with multiple cell types from blood counts, and IgA nephropathy, and CLL, as well as the IBD, PSC and eosinophil counts that are reported. These should be mentioned – as it stands the association reporting is selective. Are the associations concordant? How are the investigators sure that this locus rather than ACOXL1 is EGPA associated? Have the investigators done any of the Mendelian randomization methods such as SMR to try to pick the key gene? The same goes for the other reported loci.

These associations of variants in the *BCL2L11* region with diseases and other traits were comprehensively detailed in Supplementary Table 10 of the previously submitted manuscript, and were also shown graphically in Supplementary Figure 5A. However, the reviewer may have missed this due to errors in the indexing of the Supplementary Data which have now been corrected. Both Supplementary Table 10 and Supplementary Figure 5A detailed the associations with blood cell traits, IgA nephropathy, and CLL listed by the reviewer. At the reviewer’s suggestion, we have updated the cross-phenotype mapping using the latest information in the GWAS catalogue and eQTL databases (timestamp 22 November 2018) (revised Supplementary Table 11, plus new Supplementary Data Items 1 and 2). This exercise was helpful since there have been further GWASs published since we originally performed the cross-phenotype mapping.

We now highlight that, as had been shown in Supplementary Table 10 (now revised Supplementary Table 11), the associations in the *BCL2L11* region with IgA nephropathy, CLL and monocyte counts are not in strong LD ($r^2 > 0.6$) with the EGPA-associated variant. These signals are thus likely to be driven by a different underlying causal variant to that driving EGPA. As a result, we do not feel that these traits warrant discussion in the main text.

In contrast, the PSC-associated variant is in strong LD with the EGPA-associated variant. This is particularly relevant given the association of blood and tissue eosinophilia with PSC, and we now include discussion of this in the main text at the reviewer’s suggestion (lines 294-297).

“The EGPA risk allele is also associated with higher eosinophil count, and with increased risk of asthma, primary sclerosing cholangitis and inflammatory bowel disease (**Figure 2A, Supplementary Table 11, Supplementary Data Item 1**), diseases in which eosinophils have been implicated.”

We have updated Supplementary Table 11 and added Supplementary Data Item 1 to address the question of whether EGPA-associated variants show directionally concordant associations with other traits. For obvious reasons, this comparison can only be made for traits where the sentinel variants are actually in LD with the EGPA-associated variant (i.e. eosinophil count and PSC, and not IgA nephropathy, CLL etc.).

The reviewer asks why we favour *BCL2L11* over *ACOXL* as the likely causal gene at this locus. The reason is the strong *a priori* experimental evidence pointing to a role for BIM (the protein product of *BCL2L11*) or MORBBID in pathological eosinophilia. We do not claim that the genetic evidence *per se* favours these candidates over *ACOXL*, but clearly their known biological functions make these genes much more plausible candidates than *ACOXL*. These reasons for this distinction between *BCL2L11* and *ACOXL*, which we believe to be reasonable for the purposes of the Discussion, are supported by data presented in **Supplementary Table 12** and in response to the reviewer's comment we have added the following text (lines 292-294)

“These relevant functional associations suggest *BCL2L11* and *MORBBID* are more likely than *ACOXL* to be the causal gene at this locus.”

The reviewer proposes that we use the SMR method to try to identify the causal gene at each locus using eQTL data. Although it is becoming increasingly clear that eQTL data alone does not reliably identify the disease-causing gene (Wang and Goldstein, [biorxiv/ 459123.full.pdf](https://doi.org/10.1101/459123), 2018) we have now identified the genes at each locus for which the associated variants are eQTLs and this new data can be found in **Supplementary Data Item 2** and added the following text to the results (lines 276-278).

“In addition, we identified genes for which the sentinel EGPA-associated variants (or their proxies in LD) are eQTLs (**Supplementary Data Item 2**).”

3. How was the HLA association testing ‘adjusted for population structure’ when the main analysis suggested that with 20PCs structure effects could not be controlled (line 666-667)? In the methods it is stated that subjects of non-European ancestry were excluded. Typically the PCA is used to define ancestry and those beyond a particular limit (usually 6SDs on PC1 or PC2) are excluded. What was done here? Method should be mentioned including the PC cutoffs (so that the excluded regions can be assessed in Supplementary Figure 10). Is Supplementary 11 pre or post stratification control?

By ‘adjusted for population structure’, we meant that in the HLA association analysis, genetic principal components (PCs) were included in the logistic regression model as covariates i.e. the analysis was adjusted for PCs 1-20. As the reviewer highlights, this did not completely correct the inflated lambda value and so strictly speaking our wording was not correct. We have now revised the wording of the methods (lines 745-747) to clarify this as follows:

“Association testing for HLA variants was performed in Plink v1.9 using logistic regression including the first 20 PCs as covariates.”

The previous Supplementary Figure 10 (now Supplementary Figure 14) is before removal of non-European ancestry outliers. We have edited the Figure legend to make this clear.

“PCA plots show ancestry of EGPA patients and controls (before removal of non-European ancestry individuals) in relation to 1000 Genomes Project individuals. EGPA patients and controls are coloured dark grey. Non-European ancestry cases and controls were removed prior to subsequent analysis.”

The means and standard deviations of PC1 and PC2 were calculated for the EUR subset of 1000 genomes samples. Cases or controls lying outside +/- 3 standard deviations from the mean on either PC1 or PC2 were removed.

This detail has been added to the Methods (lines 602-606).

“Samples of non-European ancestry by PCA were excluded as described below. The means and standard deviations of PC1 and PC2 were calculated for the EUR subset of the 1000 Genomes samples. Cases or controls lying outside +/- 3 standard deviations from the mean on either PC1 or PC2 were removed.”

The QQ plots previous Supplementary Figure 11 (now Supplementary Figure 15) use unadjusted minus log₁₀ P-values. Our genomic control procedure was applied to the significance threshold rather than the p-values (so that the genome-wide significance level was set at the more stringent level of P 1x10⁻⁸ rather than 5x10⁻⁸).

*4. The HLA analysis presented in Supplementary Figure 8B suggests that HLA-DRB1*01:03 is the main-EGPA associated variant but the text lists this as the 3rd locus. Supplementary Table 12 indicates this is incorrect.*

The Figure and Table (now Supplementary Figure 11 and Supplementary Table 13) are in fact consistent with each other, and with the text. However, on reflection we can see that the data could be presented more clearly to avoid misinterpretation. For the avoidance of doubt, the primary association is at *HLA-DQA1*04:01* rather than *HLA-DRB1*01:03* (Supplementary Figure 11 and Supplementary Table 13 show this; the lowest p-value is for *HLA-DQA1*04:01*, with p 9.1x10⁻²⁰). We have re-ordered the way in which the data is presented in Supplementary Figure 11 to emphasise this. In addition, we have now modified the main text as follows to include the odds ratios for the key HLA alleles, as suggested by the reviewer (lines 381-387):

“Reciprocal stepwise conditional analysis revealed 3 signals conferred by 2 extended haplotypes encoding either HLA-DRB1*08:01-HLA-DQA1*04:01-HLA-DQB1*04:02; or HLA-DRB1*07:01-HLA-DQA1*02:01- HLA-DQB1*02:02/HLA-DQB1*03:03; together with an additional signal at HLA-DRB1*01:03 (Supplementary Table 13 and Supplementary Figure 11). The strongest independent associations with disease risk were seen at HLA-DQA1*04:01 (OR 7.18, p = 9.1x10⁻²⁰), HLA-DQA1*02:01 (OR 3.05, p = 2.0x10⁻¹¹) and HLA-DRB1*01:03 (OR 5.96, p = 2.3x10⁻⁷).”

This allele is IBD and AS-associated, both diseases that are vasculitis associated. There’s no mention of this in the Discussion. Why not – this is a massive effect so it’s a particularly strange omission?

We do not believe that associations between IBD and ankylosing spondylitis (AS) and vasculitis are relevant here. It should be stressed that by far the strongest genetic association with AS is *HLA-B27*, and not *HLA-DQ*. The reviewer’s comment here seems based on the assumption that *HLA-DRB1*01:03* had the strongest association with EGPA; we have now clarified above that *HLA-DRB1*01:03* is only the third strongest association after *HLA-DQA1*04:01* and *HLA-DQA1*02:01*.

A further point worth emphasising is that there are many sorts of vasculitis, with different anatomical distributions, genetic associations and clinical significance.

The association between IBD and vasculitis is very rare, and given that IBD is common it is not clear that the scattered reports reflect more than chance association. One recent North American study has reviewed this area (Sy A., et al. Semin Arthritis Rheum. 2016; 45:475–482), and found that the major IBD vasculitis association is with Takayasu’s arteritis, which is a large-vessel vasculitis with an MHC association distinct from that of IBD and EGPA. Of 338 patients with vasculitis and IBD in this study, only 27 had ANCA-associated vasculitis. 20 of these had GPA (predominantly associated with anti-PR3 antibodies, and thus with a distinct *HLA* association to MPO+ EGPA). Only 4 had MPA and 3 EGPA - the two groups that might contain MPO+ patients. It is hard to make much of this.

Positive ANCA immunofluorescence with a perinuclear pattern of staining (pANCA pattern) is, however, a relatively common laboratory finding in IBD (particularly in ulcerative colitis). This is rarely accompanied by vasculitis, and the autoantigen underlying pANCA staining in IBD is not MPO, which is virtually absent in ANCA+ IBD (Rozen daal & Kallenberg, Clin Exp Immunol 1999; 116: 206–213.), thus not supporting an association with *HLA-DRB1*01:03*.

As the reviewer points out an association with *HLA-DRB1*01:03* has also been reported for IBD and ankylosing spondylitis (though it is not the strongest association with EGPA). Reports of association of vasculitis and ankylosing spondylitis (AS) are very rare, and confined mainly to case reports of cutaneous leucocytoclastic vasculitis (a skin-limited small vessel vasculitis) and aortitis (a large-vessel vasculitis). Neither of these types of vasculitis are pathologically related to ANCA-associated vasculitis (a systemic small- to medium-vessel vasculitis), and they do not share known HLA associations. One study examined a large AAV cohort for co-existing inflammatory disease – AS was one of the least likely disease associations (Martín-Nares E et al. Clin Rheumatol. 2018 Jul 14. doi: 10.1007/s10067-018-4212-1). We can find only a single case report of EGPA and AS (and that was confounded by immunosuppression and systemic candidal infection).

We therefore chose not to emphasise this issue. If the Reviewer or Editor thought it helpful in case the issue was perceived to be a “relevant negative”, we could add this explanation to the Supplementary material, but we are not convinced it would be particularly illuminating.

Also, for the key HLA alleles that are associated, the ORs should be mentioned in the main text?

These have now been added to the main text (see response to point 4 above).

5. The authors make the suggestion that ANCA-negative disease may have a gut/mucosal origin rather than being autoimmune. Are they suggesting that the disease is autoreactive, or what pathological process are they suggesting? The data to support this as presented is pretty lean (although the authors have not yet apparently factored in the genetic overlap with known gut mucosal diseases such as IBD and AS). A simple test that should be done is to look at the overlap between the genetic associations and epigenetic marks in different cell and tissue types, such as using the program EPIGWAS. This could provide statistical support for what otherwise looks like a hypothesis only.

Our suggestion was a hypothesis consistent with the data, and was never intended to be a conclusive statement (e.g. in the discussion on p 14 “suggests ANCA-negative EGPA might arise as a mucosal/barrier, rather than autoimmune, disease”). As implied by the reviewer, the hypothesis was that mucosal dysfunction might contribute to pathogenesis of ANCA negative EGPA (as has been suggested for IBD, for example), rather than it being driven by autoimmunity, as is likely for MPO+ EGPA, which has evidence of autoreactivity (i.e. autoantibodies) and an *HLA* association. In response to the reviewer’s comments we have compared the genetics underpinning EGPA with IBD. This provided evidence to support our hypothesis - there was enrichment of IBD-associated variants in ANCA-negative EGPA, but no evidence of such enrichment in MPO+ EGPA. We now present QQ plots demonstrating enrichment of IBD-associated variants in EGPA (new Supplementary Figure 10) and have revised text lines 320-324 and also discussion lines 488-490, while ensuring it remains clear that this is a hypothesis rather than an established truth.

Minor Points

1. What is the contribution of the MHC vs the non-MHC loci in EGPA?

We agree with the reviewer that the relative contributions of MHC and non-MHC loci to the heritability of EGPA are interesting, and have addressed this in the MPO-ANCA +ve subset of patients in which a MHC association is seen. Using BOLT-REML to partition the variance we demonstrate that the MHC accounts for 6% of the heritable phenotypic variance and the non-MHC loci explain 94%. This observation is broadly in line with that observed for other autoimmune diseases where estimates for the MHC contribution range from 2 to 30% (Matzaraki et al, Genome Biology, 2017). We have added the following text to the results (lines 399-404).

“To quantify the relative contribution of the MHC to the heritable phenotypic variance we partitioned the variance using BOLT-REML (33). Using this approach 6% of the heritable variance could be attributed to the MHC, an observation similar to that seen in other

autoimmune diseases where the MHC contribution ranges from 2% in systemic lupus erythematosus to 30% in type 1 diabetes (34)”

2. What is the genetic variance between ANCA-positive and –negative EGPA, and if the datasets are available, with asthma?

We show that ANCA –ve EGPA is more similar to asthma than is MPO +ve EGPA (see Supplementary Fig 4 and our detailed Supplementary Note “Comparison of genetic similarity of ANCA negative”, pages 9-12 of the Supplementary Material). Due to the modest size of the EGPA subsets, it is impossible to make a reliable estimation of the genetic covariance with asthma or each other. Nevertheless, we performed an analysis of genetic correlation (rg), using LDSC (LD score regression Bulik-Sullivan et al, Nature Genetics, 2016). This showed the following:

Phenotype 1	Phenotype 2	RG	Standard error	P-value
MPO+	ANCA -	0.23	0.22	0.30
MPO+	Asthma	0.22	0.15	0.37
ANCA -	Asthma	0.40	0.20	0.055

As expected, given our modest samples sizes and thus large standard errors, the p-values are non-significant. We are thus unable to make any conclusive statement regarding the genetic covariance of these traits on the basis of this analysis. We have added the following text to the manuscript to reflect this (lines 248-250).

“Differences are also seen in the genetic relatedness to IBD (see below) but we have insufficient power to formally demonstrate genetic variance between ANCA-negative and ANCA +ve EGPA.”

3. Regarding replication, what’s the correlation of log(OR) for EGPA vs the replication dataset? High correlation supports replication (as per Figure 3a)?

We thank the reviewer for this helpful suggestion and have added a new Supplementary Figure that addresses this point (Supplementary Figure 2). This shows a strong correlation between the log(estimated ORs) for EGPA in the two cohorts (Pearson $r = 0.78$, $p = 0.003$), providing support for the reported associations. This has been reflected in the text on lines 163-166

“Moreover, there was a strong correlation between the estimated effect sizes in the primary and replication cohorts ($r = 0.78$, $p = 0.005$), providing support for the reported associations (Supplementary Figure 2).”

6. How does the data indicate that EGPA is polygenic rather than oligogenic?

We acknowledge that the numerical cut-off between oligo (few) versus poly (many) is somewhat arbitrary, but given that we identified 11 loci despite our modest sample size, we are confident that EGPA is polygenic (experience from other GWASs indicates that if larger sample sizes could be accrued, many more signals would be found). We acknowledge that in the previously submitted version we first used the phrase ‘polygenic’ at a point in the manuscript where only 3 associations had been described. We have now altered this so we first conclude EGPA is polygenic only after all 11 loci have been reported (lines 188-189).

7. BACH2 is also associated with PSC and AS 1, both diseases associated with vasculitis. Neither is mentioned. As per major point 2, is the association concordant or discordant?

We have updated the cross-phenotype mapping using the latest version of the GWAS catalogue (**Supplementary Table 11** and **Supplementary Data Item 1**); the PSC association is now listed with concordance shown (the association is discordant: the EGPA risk allele reduces PSC risk).

PSC is rarely associated with vasculitis (scattered case reports of many different sorts of vasculitis, none we can find of EGPA) and while ANCA is commonly present it is rarely directed against MPO – the relevant specificity in EGPA (Sowa PLoS One 2014;9:e107743).

The AS-associated variant is not in LD with the EGPA-associated variant; further questioning the relevance of AS to EGPA (see also HLA discussion above). Since the variants are not in LD, no comparison of effect directions is possible. As we stated, “polymorphisms in the *BACH2* region have been associated with susceptibility to numerous immune-mediated diseases”. Those traits with variants in LD with the EGPA-associated variant are asthma, nasal polyps, allergy, celiac disease, IBD, PSC and MS, rather than AS.

The results have been amended as follows (lines 346-349)

“The EGPA-associated variant is in LD with variants associated with asthma, nasal polyps, and allergy, as well as other immune-mediated diseases including celiac disease, IBD, PSC and multiple sclerosis (**Supplementary Data Item 1**).”

8. SOCS1 is associated with AS, PSC and IBD 1. This is not mentioned; all three diseases are associated with vasculitis, and IBD/PSC with ANCA. As per point 4, are the associations concordant or discordant? This information may help demonstrate that this locus is a true positive. Surely this is more relevant than ChIP data from NOD mice??

Only asthma-, PSC-, and nasal polyp-associated variants are in LD with the EGPA-associated variant. Concordance details for these traits have been added (at this locus, the directions of effect are concordant between EGPA and these 3 traits). The AS-associated variant is not in LD with the EGPA-associated variant. Please see the updated **Supplementary Table 11** and **Supplementary Data Item 1**, and earlier comments about vasculitis and ANCA associations with AS, PSC and IBD. However, nasal polyps are a common clinical feature to EGPA, and 4 of our 11 EGPA hits are associated with nasal polyps. Clearly this is relevant and is now discussed (lines 373-374).

“The EGPA-associated variant is associated with asthma, nasal polyps, and PSC”

9. Please provide the zoom plots of the reported loci in the Supplement.

In addition to the close ups of the reported genomic loci in Supplementary Figure 6, we have now included a new Supplementary Figure 7 showing plots generated with the LocusZoom software.

Reviewer #2 (Remarks to the Author):

The authors demonstrate that EGPA is polygenic with genetic distinctions between MPO+ and ANCA negative disease, correlating with different clinical features. This is well written, the tables and figures are of high quality, and discussion was clearly presented.

We are grateful to the reviewer for these comments.

Reviewer #3 (Remarks to the Author):

This paper reads very well – no major objections in terms technical flaws etc.

The figures 1 lack an x-axis and what happened to the sex chromosomes? I was wondering if there

isn't a prettier and more informative way to integrate these results into a single plot -- perhaps using different colors for the subset specific loci, and the key loci annotated? (In this way it feels like Figures 1 and 2 are very different – made by different people?) It also seems crazy in 2018 not to present the full genome-wide analysis results (including chrX).

The reviewer is correct in surmising that Figures 1 and 2 were indeed made by different co-authors, and we thank them for encouraging us to think more creatively about data visualisation. We have explored a number of options such as a “Miami” rather than “Manhattan” plot for Figure 1 but feel our original format visually demonstrates the subset-specific associations most clearly. We have corrected the x-axis omission.

In common with the majority of published GWAS studies we have not performed X or Y chromosome imputation or association testing (Wise et al American Journal of Human Genetics, 2013). Genetic analysis of the sex chromosomes comes with unique technical and statistical challenges, in part due to copy number differences between women and men, as well as the random effects of X inactivation. A robust approach to deal with the statistical issues would be to perform a sex-stratified analysis of chromosome X. However, this comes with an associated reduction in power and given the relatively small size of the cohort we were reluctant to further stratify our cohort. This is clearly an important question and as we recruit a larger cohort it should become more tractable.

On a separate note, I strongly feel the publication of the manuscript must be accompanied with public release of the summary stats of the major strata described here. There needs to be stricter enforcement from the journal.

We have included full summary statistics as **Supplementary Data Item 5**.

For figure 3 B I did not think the legend was sufficiently clear – I didn't really understand what was plotted (the numbers given). I would have found it more intuitive if the authors presented genetic correlation estimates (genome-wide) between the EGPA subsets and the traits listed.

We thank the reviewer for highlighting this. The numbers are percentages of individuals with each of the following clinical features. We have revised the figure (now **Figure 3**) and the accompanying legend (lines 1061-1069) to improve clarity.

“Figure 3. Clinically and genetically distinct subsets within EGPA, and their relation to MPO+ AAV. Above: schematic showing relationship between MPO+ AAV, MPO+ EGPA and ANCA –ve EGPA, and putative genes underlying this classification. **Below:** Unshaded cells in the table show a comparison of the clinical features of MPO+ANCA and ANCA-negative EGPA from this study as % (* p<0.0002 compared to other EGPA subset: see Table 1), but also see(4, 5). Shaded cells show data from external sources: MPO-AAV clinical data was derived from the EVGC AAV GWAS(15), and rituximab response rates for MPO-AAV from the RAVE study(59) and for EGPA from (35). n.d. = not determined”

Table 1. The table header seems more like a conclusion than a factual description of the table contents.

We have changed the title of Table 1 to “Comparison of clinical features between MPO positive and ANCA negative EGPA patients”.

Is there more clinical information wrt the (sub)type of cardiomyopathy observed in these patients?

Cardiomyopathy was demonstrated either by echocardiogram or by cardiac MRI (see **Supplementary Table 1**). Unfortunately, we do not have more detailed information available on the cardiomyopathy observed in these patients.

Spell out ENT.

We have now spelt out the abbreviation ‘ENT’ in the legend of Table 1.

The most interesting finding is the potential role of eosinophil levels – I would have loved to see an MR analysis done to test whether such levels are in fact causally driving EGPA or asthma etc?

We thank the reviewer for this helpful suggestion. We have now performed a Mendelian randomisation analysis to test whether eosinophil count is a causal factor in EGPA. Excitingly, this analysis supports a causal role for eosinophil count in EGPA. We have included these new data in the main text (lines 406-419) and figures (Figure 4B and Supplementary Figure 13).

“Mendelian randomisation supports a causal role for eosinophils in EGPA aetiology

The observation that eosinophilia is a ubiquitous clinical feature in EGPA does not necessarily imply that eosinophils play a causal role in the disease, since eosinophilia might instead be either an epiphenomenon or a downstream consequence of EGPA. The observation that 9 of the 11 alleles associated with increased EGPA risk are also associated with increased “physiological” eosinophil count supports the notion of a causal role for eosinophils in EGPA. To formally test this hypothesis, we employed the technique of Mendelian randomisation (MR) (**Methods**). In contrast to observational associations, which are liable to confounding and/or reverse causation, MR analysis is akin to a “natural” randomised trial, exploiting the random allocation of alleles at conception to allow causal inference. MR analysis provided strong support ($P < 2.8 \times 10^{-9}$, inverse-variance weighted method) for a causal effect of eosinophil count on EGPA risk (**Figure 4B**). This result was robust to a number of sensitivity analyses (**Methods, Supplementary Figure 13, Supplementary Data Item 3**)”

We have also amended the Discussion (lines 456-457).

“The causal nature of this association was supported by Mendelian randomisation.”

and the methods (lines 800-811)

“Mendelian Randomisation

“Two-sample” Mendelian randomisation (MR) was performed to assess whether there is a causal effect of eosinophil count (the exposure) on EGPA (the outcome). MR analysis was performed using the MendelianRandomization R package. Summary statistics were obtained for 209 conditionally independent variants associated with peripheral blood eosinophil count in a population study by Astle et al (13). 193 of these were typed or imputed in the EGPA dataset. MR analysis was performed using the beta coefficients and standard errors for these 193 variants in the eosinophil count GWAS and in our EGPA GWAS (all EGPA versus controls). The primary analysis was conducted using the inverse-variance weighted method. Additional sensitivity analyses were performed using alternative methods (**Supplementary Figure 13**).”

A non-reviewer driven alteration:

The log(odds ratios) in Table 2, Fig 3A, and Supplementary Figure 5 of the previously submitted version of the manuscript used the allelic ORs from the SNPTEST output, rather than the beta coefficients for the genotype term from the logistic regression model. This has now been corrected so we present ORs based on the logistic regression model (i.e. adjusted for genetic PCs 1-20).

Reviewers' Comments:

Reviewer #1:

Remarks to the Author:

Reviewer 1:

I thank the authors for their constructive approach to my suggestions. I learnt a lot about vasculitic complications of other diseases through the response! I have a few comments in regards these responses:

1. It is not clear why the authors have continued not to use results that are not properly controlled for population stratification, particularly now that they know that using LMM then can achieve appropriate levels of control. No reason is given for this approach, which is not the optimal approach, because as pointed out in my original review, using genomic control (or adjusting the Bonferroni significance threshold) assumes stratification effects across the genome are equal. I strongly advise the authors to use the LMM results throughout.

2. This also goes for the HLA analysis, where the authors are still using the 20PC approach (?the same as the original 20PCs) that result in an unacceptably high lambda. Without adequate stratification control, the HLA findings are not statistically robust. There are significant differences in case and control frequencies between the different case groups studied (Supplementary Table 14), and thus the p-values reported for the HLA associations may be influenced by stratification, and thus potentially look more robust than they may be. Given that the authors have an approach that will control properly for stratification, unless there is a good reason why not, I think it should be used.

3. It would be appropriate to quote the lambda 1000, given the case numbers are relatively low.

Minor points:

1. I apologise for not pointing to the appropriate references. However, the key AS/IBD/PSC SOCS1 SNP, rs367569, is in perfect LD $r^2=1$ with the SNP reported here (rs4781047) 1, so the rebuttal (point 7) is incorrect. The key BACH2 SNP in AS, rs72928038, is in albeit modest LD with the SNP reported here rs6454802 ($D'=0.89$, $r^2=0.28$) 1. The text covers this so no further amendment is necessary.

2. I am sure the authors will have checked this, but looking at the zoom plots, for TNP1 rs1837253 is a lone SNP association with a significance level $\sim 4-5$ log orders stronger than any other SNP in the region. I'm sure you will have checked the cluster plots and genotyping for this SNP.

1. Ellinghaus D, et al. Analysis of five chronic inflammatory diseases identifies 27 new associations and highlights disease-specific patterns at shared loci. Nat Genet 48, 510-518 (2016).

We would like to thank the reviewer for his/her further comments and suggestions regarding our manuscript. We have addressed these specific points through re-running our analyses as detailed below. Reviewer comments are in *blue italics* and changes to the text are highlighted in purple.

Reviewers' comments:

Reviewer #1 (Remarks to the Author):

I thank the authors for their constructive approach to my suggestions. I learnt a lot about vasculitic complications of other diseases through the response! I have a few comments in regards these responses:

1. It is not clear why the authors have continued not to use results that are not properly controlled for population stratification, particularly now that they know that using LMM then can achieve appropriate levels of control. No reason is given for this approach, which is not the optimal approach, because as pointed out in my original review, using genomic control (or adjusting the Bonferroni significance threshold) assumes stratification effects across the genome are equal. I strongly advise the authors to use the LMM results throughout.

At the reviewer's suggestion, we now use the results from LMM analyses throughout the manuscript. All downstream analyses that use summary statistics from the EGPA GWAS have been re-run. These analyses include the cFDR analysis, Mendelian randomisation analysis, and the overlap of EGPA-associated variants with other disease-associated variants from the GWAS catalogue. Figures 1-4 and all relevant Supplementary display and data items have been re-made based on the LMM data, and the manuscript has been revised to reflect this.

A summary of the new results and the key changes are presented below.

i) Five loci now reach genome-wide significance in the primary analysis of all EGPA vs controls using LMM. These include three loci that were previously genome-wide significant by logistic regression: the *HLA*, *BCL2L11* and *TSLP*. In addition, the intergenic region on chromosome 10 and a variant in *CDK6* are also now significant (previously these loci were significant through cFDR analysis but not logistic regression).

ii) The *SOCS1* and *TBX3* loci are no longer significant by cFDR. Variants at the *C5orf56* (near *IRF1* and *IL5*), *BACH2* and *LPP* loci remain significant by cFDR.

iii) In the analysis of MPO+ EGPA vs controls using LMM, *HLA-DQ* remains highly significant, and much more strongly so than in the analysis of all EGPA vs controls. We also detected a genome-wide significant signal in an intergenic region on chromosome 12 that was not previously apparent. As this signal was only apparent in the imputed data, and not in the directly genotyped data (see Figure 1 and Supplementary Figure 1) we have presented this new finding in a circumspect manner, highlighting this potential limitation in the Discussion.

“Finally, we highlight that variant on chromosome 12 associated specifically with the MPO+ subgroup has a low minor allele frequency and was imputed rather than directly genotyped, and so will require validation in future larger studies.”

iv) In the analysis of ANCA –ve EGPA vs controls using LMM, the *GPA33* locus remained significant. In addition, there was a signal in the *HLA* region which was not apparent from the previous logistic regression analysis. This signal is much weaker than in the MPO ANCA positive group, and further examination revealed it to be distinct from *HLA-DQ* (see new Supplementary Figure 4). The *TSLP* and *C5orf56* (near *IRF1* and *IL5*) loci also reached genome-wide significance in the ANCA –ve substrata analysis, although the *TSLP* signal appears to be independent of ANCA status.

v) The ‘within cases’ analysis of MPO positive vs ANCA negative patients using LMM revealed a genome-wide significant signal at *HLA-DQ*, as was previously detected through logistic regression. This result confirms the fundamental genetic difference between the ANCA positive and negative EGPA subsets. The chromosome 12 signal detected in the analysis of MPO+ cases versus controls was significant in the within cases analysis after Bonferroni correction (P 0.0006). The *C5orf56-IRF1-IL5* region also showed a trend towards significance. The following change was made to the text:

“This analysis of MPO+ versus ANCA –ve cases revealed two signals that were significant after Bonferroni adjustment for multiple testing (P < 0.006). There was a genome-wide significant association at rs28724235 in the *HLA* (P 2.3×10^{-15}), and an association at rs78478398 on chromosome 12 (P 0.0006). In addition, the *C5orf56-IRF1-IL5* region (rs11745587) was nominally significant (P 0.008), and only narrowly missed the multiple-testing adjusted significance threshold. There was no association identified at the other loci tested, either because of lack of power or because the signals at these loci were not subgroup-specific. Of note, the *HLA* association detected in the ANCA negative vs controls analysis was not evident in the within-cases analysis (P 0.19). In summary, this analysis provides robust evidence of a differential genetic basis of MPO+ and ANCA- EGPA at the *HLA* region.”

2. This also goes for the *HLA* analysis, where the authors are still using the 20PC approach (?the same as the original 20PCs) that result in an unacceptably high lambda. Without adequate stratification control, the *HLA* findings are not statistically robust. There are significant differences in case and control frequencies between the different case groups studied (Supplementary Table 14), and thus the p-values reported for the *HLA* associations may be influenced by stratification, and thus potentially look more robust than they may be. Given that the authors have an

approach that will control properly for stratification, unless there is a good reason why not, I think it should be used.

We thank the review for this suggestion. As suggested, we have re-run the *HLA* analysis using LMM (see revised Supplementary Figures 4 and 12). The following change was made to the text

“MPO+ANCA EGPA was associated with a region encompassing the HLA-DR and –DQ loci (Supplementary Figure 4). The classical HLA alleles at 2 or 4 digit resolution, and amino acid variants at 8 HLA loci, were then imputed. Using LMM, 9 HLA alleles conferring either susceptibility to or protection from MPO+ANCA EGPA were identified (Supplementary Table 11). Conditional analyses revealed 3 signals conferred by 2 extended haplotypes encoding either HLA-DRB1*0801-HLA-DQA1*04:01-HLA-DQB1*04:02; or HLA-DRB1*07:01-HLA-DQA1*02:01- HLA-DQB1*02:02/HLA-DQB1*03:03; together with an additional signal at HLA-DRB1*01:03 (Supplementary Table 11 and Supplementary Figure 4). The strongest independent associations with disease risk were seen at HLA-DRB1*08:01 (OR 37.6, $p = 1.3 \times 10^{-24}$), HLA-DQA1*02:01 (OR 3.2, $p = 1.2 \times 10^{-15}$) and HLA-DRB1*01:03 (OR 14.5, $p = 3.3 \times 10^{-8}$). Protection from disease was associated with the presence of HLA-DQA1*05:01 (OR 0.5, $p = 9.4 \times 10^{-10}$). A similar analysis of the MHC signal seen in the ANCA-ve EGPA subset revealed no association with any of the imputed classical alleles. Analysis of HLA allelic frequencies stratified by country of recruitment revealed a consistent pattern (Supplementary Table 12), indicating that our findings were not the result of residual population stratification.

Individual amino acid variants in HLA-DRB1, HLA-DQA1, and HLA-DQB1 were associated with MPO+ANCA EGPA (Supplementary Figure 12A). Conditioning on the most associated amino acid variants at each locus, position 74 in HLA-DRB1, position 69 in HLA-DQA1 and position 54 in HLA-DQB1 demonstrated that the three loci were independently associated with disease risk (Supplementary Figure 12B-D). Conditioning on all three variants accounted for the entire signal seen at the MHC locus. The HLA-DQ locus associated with MPO+ANCA EGPA was the same as that previously associated with MPO+ANCA vasculitis(15). To quantify the relative contribution of the MHC to the heritable phenotypic variance we partitioned the variance using BOLT-REML (30). Using this approach, 6% of the heritable variance could be attributed to the MHC, an observation similar to that seen in other autoimmune diseases where the MHC contribution ranges from 2% in systemic lupus erythematosus to 30% in type 1 diabetes (31).”

3. It would be appropriate to quote the lambda 1000, given the case numbers are relatively low.

The Lambda1000 was identical to 3 decimal places so we have not included it, however we would be happy to do so if the reviewer felt it was helpful.

Minor points:

1. I apologise for not pointing to the appropriate references. However, the key AS/IBD/PSC SOCS1 SNP, rs367569, is in perfect LD $r^2=1$ with the SNP reported here (rs4781047) 1, so the rebuttal (point 7) is incorrect.

Using the LMM analysis, the variant in the *SOCS1* region is no longer genome-wide significant by cFDR, and so this point is no longer relevant.

*The key *BACH2* SNP in AS, rs72928038, is in albeit modest LD with the SNP reported here rs6454802 ($D'=0.89$, $r^2=0.28$) 1. The text covers this so no further amendment is necessary.*

The *BACH2* variant from our new cFDR analysis remains rs6454802, so no change required.

*2. I am sure the authors will have checked this, but looking at the zoom plots, for *TNP1* rs1837253 is a lone SNP association with a significance level ~4-5 log orders stronger than any other SNP in the region. I'm sure you will have checked the cluster plots and genotyping for this SNP.*

This variant was directly genotyped – we have verified that the genotype calling is reliable. Indeed, this lone association reflects the unusual properties of this variant – it has no high LD proxies in the 1000 Genomes data. As we highlight in the manuscript,

“No SNPs are in high linkage disequilibrium (LD) with rs1837253, with no variants with $r^2 > 0.3$ in European-ancestry populations in the 1000 Genomes phase 3 data, suggesting that it is either the causal variant or, alternatively, that rs1837253 is tagging a rare variant that was not present in the individuals sequenced in the 1000 Genomes Project.”

Reviewers' Comments:

Reviewer #1:

Remarks to the Author:

I thank the authors for their constructive response to my suggestions. It's a great piece of work which I think is an important addition to the field. I'm happy with the revised paper and have no further suggestions to make.

Re: Response to reviewer for NCOMMS-18-23452B to Lyons et al “Genetically distinct clinical subsets, and associations with asthma and eosinophils abundance, with the Eosinophilic Granulomatosis with Polyangiitis (EGPA)”

REVIEWERS' COMMENTS:

Reviewer #1 (Remarks to the Author):

I thank the authors for their constructive response to my suggestions. It's a great piece of work which I think is an important addition to the field. I'm happy with the revised paper and have no further suggestions to make.

We thank the reviewer for their positive comments and feedback throughout the review process.